# Strategies of Screening and Treating Post-Extubation Dysphagia: An Overview of the Situation in Greek-Cypriot ICUs

**DOI:** 10.3390/healthcare11162283

**Published:** 2023-08-13

**Authors:** Meropi Mpouzika, Stelios Iordanou, Maria Kyranou, Katerina Iliopoulou, Stelios Parissopoulos, Maria Kalafati, Maria Karanikola, Elizabeth Papathanassoglou

**Affiliations:** 1Department of Nursing, School of Health Sciences, Cyprus University of Technology, 3041 Limassol, Cyprus; maria.kyranou@cut.ac.cy (M.K.); maria.karanikola@cut.ac.cy (M.K.); 2Limassol General Hospital, State Health Services Organization, 4131 Limassol, Cyprus; iordanou.stelios@gmail.com; 3SHPS, City University, London EC1V 4PB, UK; katerina.iliopoulou@city.ac.uk; 4Department of Nursing, University of West Attica, 12243 Athens, Greece; spariss@uniwa.gr; 5Department of Nursing, National and Kapodistrian University of Athens, 11527 Athens, Greece; mkalafat@nurs.uoa.gr; 6Faculty of Nursing, University of Alberta, Edmonton, AB T6G 2X8, Canada; papathan@ualberta.ca

**Keywords:** assessment, awareness, diagnosis, dysphagia, intensive care, management, screening, treatment, practices

## Abstract

Post-extubation dysphagia (PED) can lead to serious health problems in critically ill patients. Contrasting its high incidence rate of 12.4% reported in a recent observational study, many ICUs lack routine bedside screening, likely due to limited awareness. This study aimed to establish baseline data on the current approaches and the status of perceived best practices in PED screening and treatment, as well as to assess awareness of PED. A nationwide cross-sectional, online survey was conducted in all fourteen adult ICUs in the Republic of Cyprus in June 2018, with a 100% response rate. Over 85% of ICUs lacked a standard screening protocol for PED. The most commonly reported assessment methods were cough reflex testing and the water swallow test. Treatment approaches included muscle strengthening exercises without swallowing and swallowing exercises. Only 28.6% of ICUs acknowledged PED as a common issue. The study identified significant gaps in awareness and knowledge regarding PED screening and treatment in Greek-Cypriot ICUs. Urgent implementation of comprehensive dysphagia education programs within the units is necessary, and interdisciplinary collaboration among nurses, intensivists, and speech and language therapists is crucial to improve the quality of care provided.

## 1. Introduction

Oropharyngeal dysphagia (OD) is defined as the difficulty in transferring liquids and food from the oropharyngeal cavity into the stomach, and it refers to any abnormality in the swallowing physiology of the upper aerodigestive tract [1]. It is listed under code MD93 in the International Classification of Diseases (ICD), 11th revision [2]. In a recent systematic review and meta-analysis, the global prevalence of OD was 43.8% in different populations, while a sub-group analysis showed that it was high (48.1%) in the elderly population [3]. Researchers demonstrated that the prevalence of diagnosed OD was 3% among adult patients admitted to hospitals in the United States [4] and, in consistency with the previous results, it was higher in the population over 75 years old than in other age groups [5].

OD is a common comorbidity among various patient populations, such as stroke and cancer patients and patients with brain injury [6,7,8]. Critically ill patients are also a population at high risk to develop dysphagia as it is a significant healthcare-acquired complication due to invasive mechanical ventilation. Although the potential mechanism of dysphagia is not yet clear, it can occur after orotracheal extubation (post-extubation dysphagia, PED) even though its etiology is multifactorial [9]. Recent systematic review and meta-analysis reported that PED occurs in 41% of intubated patients [10]. A prospective observational trial identified that systematic screening for PED demonstrated a consistently high incidence rate: 12.4% after extubation in the intensive care unit (ICU) and 10.3% at ICU discharge [11]. Previous studies have also highlighted that PED may persist even after hospital discharge and can last up to six months [11,12]. Negative consequences for the ICU patients associated with PED have also been identified, including aspiration pneumonia [13], malnutrition [14], reintubation [11], short-term mortality and long-term mortality [15]. PED also prolongs ICU and hospital length of stay (LOS) [14] and increases resources use [16]. Additionally, it increases the risk of poor outcomes in ICU survivors as it has a direct negative impact on their quality of life [17] and independence [18]. Therefore, it appears from the above that PED remains a critical issue amongst critically il patients.

Despite PED’s high prevalence and significant association with negative patient outcomes, bedside screening is not routinely conducted [12]. Limited awareness and inadequate knowledge of healthcare professionals [19], especially of nurses [20,21,22] but also of physicians [23], are some of the reasons that were attributed to limited screening for PED in the intensive care units (ICUs). Additionally, although a speech and language pathologist/therapist (SLP/SLT) is helpful in the evaluation and management of dysphagia [24], dysphagia is a complex problem that requires a multi-professional targeted approach [22,25,26]. Moreover, little evidence showed that assessment and management of dysphagia are conducted by nurses and non-specialists in ICUs [27].

Recent screening studies in ICUs reported that patients with frailty represent approximately 30% of critically ill patients [28,29]. Frailty, among other things, is associated with prolonged ICU stay and mechanical ventilation [28,29,30]. Furthermore, as there has been an increase in life expectancy globally, a higher number of elderly patients will be admitted to the ICUs. Given that dysphagia has been associated with mechanical ventilation, age and frailty [31,32], it is apparent that dysphagia is a critical area of concern in ICU patients. But these data will only bring value if they contribute to the way we provide care. Based on this, PED awareness is an important factor in screening, early diagnosis and treatment of dysphagia, as early identification is positively associated with treatment interventions [33].

Within the specific context of Cyprus, there exists a significant research gap regarding the prevalence of PED. The absence of prior identification and understanding of PED in Cypriot ICUs represents a substantial knowledge deficiency that needs to be addressed. By investigating the prevalence of PED in Greek-Cypriot ICUs, we can contribute to a more comprehensive understanding of dysphagia in the ICU setting and underscore the importance of investigating and addressing PED prevalence in Greek-Cypriot ICUs. Moreover, even though there are speech language therapists who are actively involved in the OD assessment and treatment in the community, there are not known protocols regarding assessment and treatment of dysphagic patients during their hospitalization in ICUs. We conducted this study to establish current approaches of Greek Cypriot ICUs to PED screening, management and treatment, as well as their current status of perceived best practices. We also aimed to assess ICU awareness of PED and its consequences.

## 2. Materials and Methods

### 2.1. Design and Settings

We used a cross-sectional online survey design, as part of the international survey titled “Dysphagia in Intensive Care Evaluation (DICE)” in which 26 countries, including Cyprus, participated [27]. The study was conducted in all 14 adult ICUs in the Republic of Cyprus. A Google Forms link to the survey was emailed by the principal investigator to the participating ICUs’ senior nurses in June 2018. Only two email reminders were sent to non-responder nurses of four ICUs at one-week intervals.

Strengthening the Reporting of Observational Studies in Epidemiology (STROBE) was followed [34].

### 2.2. Participants

According to the original study (DICE) [27], the participants were all the fourteen adult ICUs in the Republic of Cyprus designated for the management of critically ill, intubated patients. This was the final sample size of the study as there were no exclusion criteria. The Cypriot national coordinator of the study was responsible, through her personal and professional network, to recruit one senior nurse from each ICU with a good command of the Greek language to complete the questionnaire. Prior to the release of the survey, senior nurses were contacted by phone by the Cypriot national coordinator herself. They were asked, before completing the questionnaire, in order to enhance the accuracy of the data provided, to meet and collectively discuss the responses with the ICU interprofessional team involved in the assessment and treatment of dysphagia. Based on this, a questionnaire was completed by each ICU and expressed the everyday clinical practice in the particular ICU. The ICU teams involved nurses, intensivists, physiotherapists and SLP/SLTs, where available.

### 2.3. Data Collection and Instrument

We used the anonymous self-administered survey which was used in the international survey of Spronk et al. (2022) [27]. At the beginning of the questionnaire, there was a cover letter describing the study and its objectives in detail. The 46-question survey sheet contained five different types of questions: 7-point Likert scale, multiple-choice, checkboxes, matrix as well as open-ended questions that required short answers, including the following: Demographics

Seven questions (5 multiple choice and 2 checkboxes) assessing the ICU demographics.

Domain 1: Current practice

16 items assessing current ICU practices on:(a)PED management with questions (7 multiple choice, 3 checkboxes and 1 matrix) about the existing protocols for screening, methods used to confirm the presence of PED and responsibilities of every ICU team member for assessing PED.(b)Prevention of aspiration pneumonia related to PED (2 matrix questions).(c)PED treatment interventions (1 matrix, 1 checkbox and 1 multiple choice question).

Domain 2: Scope of the Problem

Ten questions (seven 7-point Likert scale and 3 multiple choice ones) were used to assess the awareness of PED and its consequences. Regarding Likert scale questions, five of them ranged from 1—Strongly disagree to 7—Strongly agree and two from 1—Strongly agree to 7—Strongly disagree.

Domain 3: Perceived Best Practice

Thirteen questions (6 checkboxes, 4 open-ended and three 7-point Likert scale questions) assessed the perception of best practices in screening and treating PED. In this domain, two of Likert scale questions ranged from 1—Strongly disagree to 7—Strongly agree and one ranged from 1—Strongly agree to 7—Strongly disagree.

As described above, three out of ten Likert scale questions were inverted, following the questionnaire creator’s advice, to avoid response bias. Finally, the terms dysphagia, OD and PED were used interchangeably in the questionnaire, as appropriate to the question.

#### Translation and Cultural Adaptation of the Instrument

In this study, we adopted the Report of the International Society for Pharmacoeconomics and Outcomes Research (ISPOR) Task Force for Translation and Cultural Adaptation [35]. First, the questionnaire was translated from English to Greek by two independent Greek, bilingual academics, one specialized in evidenced-based practice and the other in critical care nursing. The Cypriot national coordinator of the study collected and combined the two versions, and a third Greek translation of the questionnaire was obtained. Then, another independent bilingual academic specialized in teaching and learning methodology in nursing translated the Greek version back into English without having read the original English version of the questionnaire. Finally, the Cypriot national coordinator of the study compared the original English version with the back-translated English version. To assess the face validity of the translated questionnaire, it was reviewed by an external panel of experts consisting of two academics with ICU experience for more than 10 years and four post-graduate nursing students: two Ph.D. students and two master’s students, all familiar with the Cypriot ICU context. None of the reviewers reported anything that was ambiguous or hard to comprehend. The Cronbach’s alpha for the translated questionnaire was 0.96.

### 2.4. Ethics Approval

The study protocol was approved by the National Committee of Bioethics of Cyprus (EEBK/EP/2018.01.99). Submission of a completed questionnaire was considered as participation after informed consent. Anonymity was assured as neither the identity of the senior nurse who completed the questionnaire nor the identity of any person of the ICU team were collected at any point of the study. Additionally, on the last line of the study’s cover letter was written that all data would remain deidentified, be only accessible to the research team and would be securely stored in password-protected files. No intervention to patients was carried out during the study.

### 2.5. Data Analysis

Google forms were used to collect data. The data were extracted in an Excel spreadsheet, quality-tested, and analyzed. Data were summarized using descriptive statistics. Likert scale results were presented as means (M), medians, interquartile range (IQR), modal values and standard deviations (SD). Agreement with a specific statement in the questionnaire was defined as a score of 5–7 on a Likert-7 scale, while 4 was rated as neutral, and 1–3 was rated as disagreement. For the statistical analysis, the results of the three inverted Likert scale questions were reversed back (1—Strongly Disagree and 7—Strongly Agree). 

Categorical variables such as the questions regarding assessment methods were summarized as counts and percentages. Based on the design and structure of the questionnaire along with the limited surveyed population, our analysis was restricted to descriptive statistics only.

Finally, the four open-type questions were not answered; therefore, they were not included in the analysis.

The analysis was performed using the Statistical Package for the Social Sciences (SPSS) version 21. A non-parametric Spearman correlation was used to assess the relationship between public and private ICU data, and Excel (Microsoft Corporation, Redmond, WA, USA) was used to draw the graphs.

## 3. Results

Fourteen ICUs across the Republic of Cyprus participated in the survey, with a response rate of 100%. Eight ICUs had a capacity of 5–9 beds, four had 10–14, and two 15–19. Eight ICUs were located in hospitals with a capacity of fewer than 200 beds and six in hospitals with 200–499 beds. Twelve ICUs admitted patients with mixed medical and surgical problems, while four of them could also treat neurosurgical and three cardiothoracic patients. One ICU was a dedicated coronary unit and another a burn unit. Of the ICUs, five were located in private hospitals while nine were in public ones. None of the ICUs had dedicated an SLP/SLT for the needs of their patients, eight ICUs had access to an SLP/SLT upon request, and six did not have access at all (Table 1). In all ICU interprofessional teams, there was an intensivist, and in those that had access to an SLP/SLT, he also took part in it.

### 3.1. Current Practices on PED

#### 3.1.1. PED Management

##### Existing Protocols and Subgroups of Patients Screened

More than 85% of ICU teams (12 ICUs) reported that there was no standard protocol indicating which patients should be screened for PED.

Nine ICU teams reported that no patient was screened for PED in their ICU. Two ICU teams reported that 51–75% of their patients screened for PED, and in one ICU the percentage was 25–50% of patients. Screening for PED for less than 25% of their patients was reported in one ICU, and only one ICU team reported that more than 75% of their patients screened for PED.

Six ICU teams reported that more than 75% of their patients who received a tracheostomy during their ICU admission screened for PED. Three ICU teams reported that 51–75% of their tracheotomized patients screened for PED, in three ICUs the percentage was 25–50% of their patients and another three teams reported that no patient with tracheostomy was screened for PED in their ICU. Screening for PED after tracheostomy for less than 25% of patients was reported in one ICU.

Screening for PED primarily took place for those patients who demonstrated signs of dysphagia, followed by patients with existing neurological disorders and traumatic brain injury, while patients with tracheostomy came third. Subgroups of patients routinely screened for dysphagia after their extubation are presented in Figure 1.

##### Timing of Screening

After ICU admission

Dysphagia screening on ICU admission was only reported by one unit (7.1%). One ICU team reported the time of screening to be between 3 and 7 days after admission, in two units (14.2%) screening took place within 24–48 h after admission and one unit commented that they checked for dysphagia only when it was noticed upon admission. No screening took place in nine ICUs (64.2%).

After extubation

Regarding the first screening for PED, no screening took place in three ICUs (21.4%). Four ICU teams (28.5%) reported that screening happened on the day of extubation, and another four that this happened 3 to 7 days after extubation. In one ICU, the screening took place 24 to 48 h post-extubation. Two ICUs (14.2%) commented that screening happened after extubation only if dysphagia was noticed.

##### Methods Used for PED Assessment

Regarding the most commonly used method in more than 75% of ICU patients to confirm the presence of PED, cough reflex testing was chosen by six ICUs (42.8%), followed by the water swallow test (5 ICUs, 35%), while oral mechanism exam, volume–viscosity swallow test (V-VST) and methylene blue test were reported in three ICU teams, respectively. Videofluoroscopic swallowing study (VFSS), flexible endoscopic evaluation of swallowing (FEES) and V-VST were not used at all in seven ICUs (50%), the self-reported method was not used in six units, and oral mechanism exam as well as gugging swallowing screen (GUSS) were not used in five ICUs (35.7%), respectively. Three ICU teams reported that they were unfamiliar with VFSS, cervical auscultation, V-VST and oral mechanism exam. FEES was ranked the lowest among all methods since it was used only in 25% of patients by one ICU, and VFSS ranked second since it was used as assessing method in more than 75% of patients by only one ICU. The vast majority of the participants reported that FEES and VFSS were not used, were not available or the healthcare professionals were unfamiliar with them (Figure 2).

##### Responsibilities of ICU Team Members

In the majority of ICUs, nurses and intensivists were mostly responsible to assess PED as well as SLP/SLTs (Figure 3). Eight ICU teams (57.1%) reported that, when an SLP/SLT was available, the percentage of patients consulted was <25%. The availability of an otolaryngologist/ENT specialist in order to consult when PED was suspected was reported in four ICUs (28.6%), and the percentage of patients consulted was <25% in three ICUs (21.4%). Only one ICU team reported a proportion of patients consulting otolaryngologist/ENT specialist of >75%. Figure 4 presents the responsibility of each ICU member in screening methods used for PED.

#### 3.1.2. Prevention of Aspiration Pneumonia Related to PED

##### Aspiration/Aspiration Pneumonia Resulting from Liquids/Solid Food

Postural adjustment measures for the prevention of aspiration pneumonia in all patients were ranked as the most common interventions (11 ICUs, 78.6%), oral hygiene as second-most common (10 ICUs, 71.4%), delayed feeding as third (9 ICUs, 64.2%), and tracheostomy cuff deflation during meals as fourth (8 ICUs, 57.1%).

Percutaneous endoscopic gastrostomy (PEG) was the least used aspiration prevention method since the vast majority of ICU teams (seven ICUs, 50%) reported that usage was less than 25% of patients. Three ICUs (21.4%) used PEG between 25–50% of patients, in one ICU it was not used, and in one the healthcare professionals were unfamiliar with the procedure. Measures taken to prevent aspiration/aspiration pneumonia are listed in Figure 5.

##### Aspiration Pneumonia Resulting from Saliva Production

The only saliva production interventions that were used in the ICUs of the study were per-os (<25% of patients in three ICUs, 25–50% in one ICU and >75% in two ICUs) or intravenous (<25% of patients in three ICUs, 25–50% in two ICUs and >75% in one ICU) anticholinergics (e.g., glycopyrronium, atropine). Scopolamine patches, botulinum toxin type A and irradiation of the salivary gland procedures were not used.

#### 3.1.3. Interventions to Treat PED

The most commonly used interventions to treat PED were muscle strengthening exercises without swallowing and repetitive swallowing exercises/maneuvers with or without additional resistance followed by respiratory exercises. The remaining interventions, including muscle-strengthening exercises using apps on a tablet/iPad, neuromuscular electrical stimulation (NEMS) of swallowing muscles, surface EMG (sEMG) biofeedback swallowing training and pharyngeal electrical stimulation (PES) were mostly not used, or the personnel was unfamiliar with them (Table 2).

Regarding bedside swallow training, in the majority of ICUs, SLP/SLTs and nurses were identified as the team members who performed the bedside patient swallow training, followed by intensivists and physiotherapists. ENT was the member with the least bedside patient training (Figure 6). Six ICU teams mentioned that bedside teaching was not performed in their unit.

### 3.2. Scope of the Problem

#### 3.2.1. Awareness of PED Incidence

Six ICUs (42.8%%) did not agree that PED was common in their unit, whereas four ICUs (28.5%) agreed. The remaining four ICUs gave a neutral answer [mean: 3.35; SD: 1.19, Median: 4, IQR: 3, Modal Value: 5 (6)]. Ten units (71.4%) agreed that PED was associated with the duration of intubation. Five ICUs (35.7%) indicated that PED occurred in less than 25% of their ICU patients who remained intubated for more than 48 h, while another five ICU teams estimated that it occurred in 25–50% and one estimated that it occurred in between 51–75%. On the contrary, three ICU teams (21.4%) estimated that none of their patients who remained intubated for >48 h but less than 7 days developed PED. As per the estimation of how common PED was for patients who remained intubated for more than 7 days, six ICU teams (42.8%) estimated the incidence of PED as 25–50%, four estimated as less than 25%, while three ICU teams estimated it between 51 to 75%. One ICU team estimated 0% of PED in their patients who remained intubated for more than 7 days. For critically ill patients who received a tracheostomy during their hospitalization in the ICU, six ICU teams estimated that PED occurred in 51–75% of those patients, one in >75%, four in 25 to 50%, two in less than 25%, and one team estimated that no PED occurred.

#### 3.2.2. Awareness of PED Consequences

All ICU teams agreed that PED increases ICU and hospital LOS, delays patients’ return to normal functioning and influences the need for long-term care. Awareness of PED consequences are listed in Table 3.

### 3.3. Perceived Best Practices on PED

#### 3.3.1. Protocols and Routine Screening

In total, 71.4% of ICU teams [mean: 5.21; SD: 2.5; median: 7; IQR: 5; modal value: 7 (4)] agreed that a standard protocol or algorithm should be used for PED screening. Also, 71.4% of them [mean: 5.14; SD: 1.87, median: 5.5; IQR: 3; modal value: 7 (4)] identified the need for routine PED screening before the discharge of patients with a length of ICU stay of more than 48 h and the same percentage [71.4%; mean: 5.35; SD: 1.39; median: 6; IQR: 2; modal value: 6 (5)] agreed that patients who remained intubated for more than 48 h should be routinely screened for dysphagia.

#### 3.3.2. Availability of Screening and Treating Methods

Regarding ICU teams’ opinion about which screening method should be available for PED (Table 4), they chose all the possible answers as there was no limitation to the number of answers they could choose, as they did in the next question regarding their opinion about which method should be available for the treatment of PED (Table 5). The only answer that was not chosen in this question was the one stating that there is “No need for dysphagia-specific treatment”, as dysphagia will disappear when the patient’s strength increases.

#### 3.3.3. Barriers to Standardized Screening and Treatment

More than 92% of ICUs (13/14) identified the lack of protocols regarding screening and treating of PED and specialized personnel as important barriers for the implementation of standardized screening and treatment. Additionally, they recognized the lack of knowledge and the lack of education on possible treatments of PED as extra barriers for standardized screening and treatment, respectively.

#### 3.3.4. Facilitators to Standardized Screening and Treatment

As regards to the most important facilitators to standardized screening and treatment, more than 85% of ICU teams (12 ICUS) chose the use of a standardized protocol, the availability of specially trained personnel and the knowledge and ability of the ICU members to identify and treat PED.

## 4. Discussion

We conducted a survey involving health care teams working in all ICUs (14/14) in the Republic of Cyprus to assess awareness of PED and its consequences and to explore what the ICU teams perceived as best practices, as well as current approaches to PED assessment and treatment. The survey was part of DICE, a multi-center, international, online cross-sectional survey which aimed to provide evidence-based guidance for the implementation of OD protocols [27].

The interpretation of the findings follows the same categorization as in the Results section. Overall, our results show that a few ICU teams in Cyprus were aware of PED incidence in their units and most of them were aware of PED complications. Despite recognition of the need for evidence-based protocols as best practices for the screening and treatment of PED by most ICUs, very few routinely screened for dysphagia using appropriate methods. Similarly, protocols to guide PED management were not used in most ICUs, and effective treatments were limited by the lack of SLP/SPTs and/or knowledge gaps in ICU interprofessional teams.

### 4.1. Current Practices on PED

#### 4.1.1. PED Management

##### Existing Protocols and Subgroups of Patients Screened

Most of the ICU teams in Cyprus reported the absence of any standardized dysphagia assessment protocol. The absence of an assessment protocol and screening procedures in the ICU is commonly reported by most investigators [23,36,37,38]. The percentage of ICU teams in Cyprus screening for dysphagia after tracheostomy is slightly improved and similar to practices in other countries [36,37], probably due to the perceived vulnerability of this population for dysphagia. The implications of not having a standard protocol for PED screening and the potential impact on patient outcomes [10,11] point to an urgent need for the development of international guidelines for the screening and management of PED dysphagia. Based on the guidelines, new educational programs can be designed and implemented across countries to assure safe clinical practice. At the same time, the guidelines need to stress the necessity for interdisciplinary collaboration between ICU staff and SLP/SLTs due to the high level of specialization required for the management of PED and the unsettling consequences for patients if left untreated [39]. Yet, to address the limited availability of SLP/SLTs in ICUs in Cyprus, organizational choices are required at the level of health policy.

##### Timing of Screening

After ICU admission

Screening for dysphagia in the early stages of ICU admission is rarely practiced in Cyprus [no screening was reported by nine ICU teams (64.2%)]. Even when it does take place, there is variation in the timing ranging from 24–48 h (in 2 ICUs) to 3 to 7 days after admission (in one ICU). These findings are consistent with the lack of screening at admission reported in the literature [40] and probably reflect the lack of appreciation for the varying risk of dysphagia in different ICU patient groups.

After extubation

The same percentage of ICU teams reported that PED screening was performed at the day of extubation (28.5%) as well as 3 to 7 days after extubation (28.5%). No screening took place in three ICUs (21.4%). However, data from eight ICUs in Japan [41] found significant associations between each day of post-extubation delay in SLT consultation and dysphagia, aspiration pneumonia or death at the 7th, 14th, or 28th day after extubation. As a result, it is critical to appreciate timely post-extubation evaluations by trained professionals in order to implement timely interventions and prevent serious complications in high-risk ICU patients.

##### Methods Used for PED Assessment

For the limited number of ICU teams in Cyprus which reported screening for PED, the most frequent screening method was cough reflex testing followed by the water swallow test. The current practice is controversial in terms of accuracy in detecting PED and may contribute to silent aspiration and consequently pneumonia [27,42]. Instrumental assessment methods such as FEES or VFSS were rarely used in Cyprus. Similar patterns of practice were reported by other studies both for the predominance of water swallow testing and the lack of use of high-accuracy detection methods [23,27,36,37,43]. Nevertheless, comprehensive and instrumental assessments, such as VFSS and FEES, are necessary for patients in the presence of clinical signs of aspiration when water swallow screening is negative [38]. The observed paucity in their clinical application is attributed to the need for trained professionals and the availability of technological equipment which are mostly available at university hospitals [27].

##### Responsibilities of ICU Team Members

In the majority of ICUs in our study, nurses and intensivists were responsible to assess PED as well as SLP/SLTs, whereas in Switzerland nurses had the lead in the initial ICU dysphagia screening [9] and physicians in Germany [23]. None of the ICU teams that participated in our survey reported a dedicated SLP/SLT for ICU patients, while approximately half of them reported a lack of SLPs/SLTs even as an external partner. In case of an available SLP/SLT on request, the percentage of patients consulted was less than 25%. It would be very interesting to investigate how many SLPs/SLTs are available at the moment in Cyprus for ICU consultations including PED management. Although the collaboration with an SLP/SPT in the ICU can positively affect ICU-related complications such as PED [44], the lack of SLP/SLTs involved in ICU patient care is common practice across the world [45].

What complicates the situation is that, according to the American Speech-Language-Hearing Association (ASHA), SLPs are the most qualified providers for dysphagia services and “cross-training of clinical skills is not appropriate at the professional level of practice” [46]. Yet, in some countries SLPs do not receive ICU-specific training, which may explain the tradition for lack of ICU-dedicated SLPs [47]. In the absence of SLP/SLTs, PED identification has traditionally been performed by other healthcare professionals, mainly nurses. Nurse-performed dysphagia screening is considered to be feasible [26], safe [48] and superior to no screening in terms of patient outcome [49]. Until a dedicated SLP/SLT for PED screening and treatment becomes available for all ICUs, the empowerment of nurses through education along with the implementation of standardized protocols can contribute to the early identification of high-risk individuals for dysphagia and lead to referrals for optimal management. Apparently, professional and regulatory bodies of different health care professionals need to promote interdisciplinary collaboration early in the education of undergraduate students, which will hopefully lead to collaboration during clinical practice having patients as the point of reference.

#### 4.1.2. Prevention of Aspiration Pneumonia Related to PED

Postural adjustment, as well as oral hygiene, were the most widely used methods to decrease the risk of aspiration after suspected or confirmed PED in our study.

##### Aspiration/Aspiration Pneumonia Resulting from Liquids/Solid Food

Postural adjustment has been proven to promote swallowing in patients with confirmed or suspected dysphagia by affecting bolus flow and speed, especially when the patient has been placed in a sitting position [50]. Importantly, irrespective of the bolus volume, manipulating the cervical and shoulder angle has been shown to activate more effectively swallowing-related muscles during thoracic upright sitting [51].

It is established that the maintenance of good oral hygiene decreases the risk of aspiration pneumonia in the ICU [52,53]. Although dysphagia is a recognized as a risk factor for aspiration pneumonia, it is speculated that it contributes to its causation in combination with other risk factors such as poor oral hygiene [54]. As such, systematic oral hygiene can address the bacterial colonization of the oropharyngeal cavity and decrease the risk for dysphagia [55,56,57]. Furthermore, there is evidence that the cough reflex is improved with regular oral hygiene, which can act synergistically in the reduction of aspiration risk [58].

##### Aspiration Pneumonia Resulting from Saliva Production

Hypersalivation poses a serious aspiration risk for individuals with dysphagia since the normal clearance of secretions is impaired. The restricted use of saliva management interventions by the ICU teams in our study probably depicts the lack of evidence in the published literature specifically for critically ill patients [59]. Similarly, hypersalivation due to swallowing difficulties has diverse aetiologies, and multidisciplinary collaboration is required to identify the causes and implement appropriate treatment.

The major challenge in implementing interventions for the prevention of aspiration pneumonia related to PED consistently across ICUs is the lack of studies on the topic specifically for critically ill patients. Focused research could try and replicate proven interventions from other populations in critical care and/or explore both new pharmacological and non-pharmacological management options.

#### 4.1.3. Interventions to Treat PED

Muscle-strengthening exercises without swallowing and repetitive swallowing exercises/maneuvers with or without additional resistance have been identified as the most widely used interventions to treat PED in our survey. Still, they were only used by a limited number of participating ICUs. Although there is proof that these exercises promote muscle strengthening [60,61,62] and consequently swallowing, recent advances in post-extubation therapy employ swallowing techniques aided by surface electromyography [63] as well as electrostimulation of the pharynx for dysphagia treatment [64,65] with promising results. The ICU teams in our study were not familiar with these new treatment modalities, a finding that needs further consideration. Certainly, introducing these approaches requires a comprehensive approach involving targeted education and multidisciplinary collaboration, which were identified mostly as unavailable in the ICUs of Cyprus.

### 4.2. Scope of the Problem

#### 4.2.1. Awareness of PED Incidence

Only 28.5% of the participating ICU teams recognized PED as common amongst ICU patients, which suggests low awareness of dysphagia in the participating ICU teams. Our results are very similar to the findings reported by the Swiss survey of dysphagia care [43], the MAD-ICU study in Germany [23] and a Dutch national ICU survey [36] but are lower than the frequency of OD occurrence reported in the DICE international study (47%) [27]. Many respondents in our survey thought that the duration of intubation and the presence of tracheostomy increase the PED occurrence, which has also been demonstrated in the literature [9,66,67]. In 42.8% of the ICU teams, the incidence of PED was estimated as 25–50% for patients who remained intubated for more than 7 days, while in 21.4% of ICU teams, between 51 to 75%, which was less than the estimation of the DICE study (67%) [27]. The incidence of PED in patients with a tracheostomy was estimated to be 51–75% by most respondents in our study, with 25–50% being the second most frequent estimate. This is in accordance with the Dutch study [36] with cohorts including non-neurologic critically ill patients with a tracheostomy [68] as well as neurologic patients [69,70].

#### 4.2.2. Awareness of PED Consequences

The vast majority of the participating ICUs in our study agreed that the duration of ICU stay was associated with increased PED occurrence. Yet, the reasons for the prolonged ICU stay were not known since no scoring system was used in the current study. However, there is evidence that ICU patients’ disease severity in the Republic of Cyprus is high [52]. As patients with increased disease severity stay longer in the ICU and have a longer duration of intubation than others with less severe conditions [71,72], thus, they are more likely to develop PED [72]. Additionally, it is well evidenced that PED patients have a significantly longer LOS in hospitals in comparison to patients with normal swallowing [73,74,75], a finding that more than two-thirds of the participating ICUs in our study agreed with. The ICU teams in Cyprus seemed more aware of the contribution of dysphagia in the prolongation of ICU and hospital LOS compared to the findings of the DICE study [27]. Yet, it remains unclear whether PED resulted in the increased LOS or whether the increased LOS resulted in PED in our participants’ reports. A prospective observational study could help answer this question.

The functional status of a person is defined as the ability to carry out daily living abilities including eating [76]. Since PED has the potential to compromise eating ability in hospitalized patients, it can delay the return of a patient to pre-hospitalization functioning status, a finding that was also reported by all the participating ICUs of this study. Additionally, all participants agreed that patients who present with PED may need long-term care in comparison with patients without. Furthermore, in our study, almost all of the participating ICUs perceived that ICU readmission was associated with PED. Although, the association seems very possible, it has not been positively correlated among patients with clinically significant PED compared to those without [77]. 

The observed variation in the estimates of PED incidence and consequences between the ICU teams in Cyprus might reflect differences between the institutions in terms of patients’ illness category and severity. However, it can also be an indication of the absence of available SLPs/SLTs for specialized consultation in ICUs in Cyprus.

### 4.3. Perceived Best Practices on PED

#### 4.3.1. Protocols and Routine Screening

A large percentage of participating ICUs agreed that a standard protocol should be used for PED screening for patients who remained in the ICU and/or remained intubated longer than 48 h. In combination with the absence of protocols for PED screening and management currently in Cyprus, this is encouraging for future efforts towards evidence-based practices. It might signify that ICU teams are ready to welcome guidelines as soon as they are developed and implement them diligently at a national level.

#### 4.3.2. Availability of Screening and Treating Methods

Similarly, the fact that ICU teams in our study favored all PED screening and treating methods potentially attests to the level of readiness of ICU teams to adopt new practices.

#### 4.3.3. Barriers to Standardized Screening and Treatment

The ICU teams of Cyprus seem to be conscious of the lack of evidence-based protocols for PED screening and treatment since they recognize it as the most important barrier in the implementation of standardized PED management. Evidence-based practice is strongly connected to the improvement of patient outcomes [78], and the importance of protocols for PED has already been emphasized. What merits further exploration is the association between the implementation of new protocols in naïve environments and the improvement of patient-reported outcomes [79]. Additionally, educational institutions along with professional bodies need to employ research results to design courses that could address the educational gap identified by ICU teams for PED screening and treatment.

#### 4.3.4. Facilitators to Standardized Screening and Treatment

What was seen as a barrier for standardized PED screening and treatment (lack of protocols and education) was reported as a facilitator if addressed at an organizational level. Interestingly, the collaboration with specially trained personnel was only reported as a facilitator, possibly implying the complete unavailability of SLPs/SLTs and the anticipation by ICU teams to improve the care they offer to patients in the context of multidisciplinary collaboration.

## 5. Limitations

The current survey sought to map the state of post-extubation dysphagia management in critical care in the Republic of Cyprus; however, certain limitations, mostly associated with sampling methodology and size, must be acknowledged. 

First, this study presents a secondary analysis of an international cross-sectional survey conducted in 26 countries [27]. So, an important limitation of the present work regards the fact that the findings are reported in relation to a particular healthcare setting, i.e., ICUs in the public and private sector of the Republic of Cyprus. It is important to note that ICU nurses in different healthcare systems across nations may implement different protocols and care plans, or they may receive to various degrees continuing education on clinical guidelines. These concerns may influence the generalizability of the present results. However, the strength of the present study is the use of a nationwide representative sample of ICU nurses with a response rate of 100%, which partially increases the external validity of the study. Moreover, the present findings are based on a robust study design, which further supports the internal validity of the study. Additionally, we still believe that our findings can provide a contribution to the existing literature as the context and environment of care can vary significantly among countries. Based on the above, regarding the results from Greek-Cypriot ICUs, this manuscript presented differences that are discussed in the discussion section compared to the findings of the DICE study [e.g., the ICU teams in Cyprus seemed more aware of the risk of dysphagia to increase ICU and hospital LOS (85.7% and 71.4% of respondents respectively) compared to the findings of the DICE study (64% and 42% respectively)]. Additionally, the results presented are more detailed. Finally, as, to the best of our knowledge, there is a small number of nationwide studies about the management of post-extubation dysphagia and awareness of best practices in ICUs (e.g., Swiss survey of dysphagia care [43], the MADICU study in Germany [23] and a Dutch national ICU survey [36]), reporting results from individual countries is important in informing quality improvement and education efforts. Not publishing these results would imply similar trends in a specific country, when in fact this is not the case.

Second, since the results were based on the perceptions of ICU teams around the incidence, risks and management of dysphagia in their units, the findings may not accurately reflect the actual prevalence and practices related to dysphagia management in critically ill patients. 

Third, we did not collect data on the actual incidence and management of post-extubation dysphagia, as this could only be done prospectively due to lack of consistent documentation. This limitation implies that this study’s findings may not capture the true scope and characteristics of dysphagia in critically ill patients. 

Suggesting directions for future research, a prospective data collection, after ensuring that consistent and comprehensive documentation practices are in place to facilitate data collection, will provide more accurate and reliable information compared to relying on perceptions. Combining quantitative research with qualitative research methods such as interviews or focus groups involving multiple ICUs across different regions or countries will also help to enhance the generalizability of the findings and provide a more comprehensive understanding of dysphagia management practices in different healthcare settings.

Fourth, the absence of the national study coordinator during the questionnaire completion introduces the possibility of response bias. While senior nurses verified the completed questionnaires with the ICU team members, the lack of direct oversight raises concerns about the accuracy and consistency of responses. To minimize response bias in future research, having a study coordinator or research assistant present during the completion of questionnaires can ensure that the questions are understood correctly and provide an opportunity for immediate clarification if needed.

Finally, although we checked internal reliability and face validity of the questionnaire used, several other forms of reliability and validity were not performed, e.g., test-retest analysis, concurrent validity, etc. This limitation suggests that the robustness and accuracy of the study’s findings may be compromised. Future research should include test-retest analysis to evaluate the stability of the questionnaire over time and concurrent validity testing to establish its relationship with other relevant measures.

Overall, the limitations reported include the secondary analysis, the perception-based results, the lack of data on actual incidence and management, the response bias and questionnaire completion, and the lack of comprehensive reliability and validity testing. These limitations highlight the need for caution when generalizing the study findings on the management of dysphagia in critically ill patients, especially to the post-COVID era where the landscape of critical care and ICU practices may have evolved and new guidelines, interventions or technologies may have emerged. In addition to the above suggestions for future research, further research, considering the unique challenges and changes brought about by the COVID-19 pandemic, may be necessary to provide more reliable and applicable insights into dysphagia management in critically ill patients in the post-COVID era, leading to improved patient outcomes and quality of care. 

## 6. Conclusions

According to the results of our study, ICU teams in Cyprus demonstrated low levels of awareness and knowledge regarding PED. Although perceived best practices were identified, there were no established protocols for the management of dysphagia. As a consequence of high disease severity of ICU patients in the Republic of Cyprus [52], longer ICU hospitalizations are expected, which in turn are associated with the occurrence of dysphagia [52,72,73]. 

The findings for PED screening and treatment in Cyprus in combination with those by other countries are unsettling for the care of critically ill patients in ICUs. In the context of the COVID-19 pandemic and the associated respiratory failure and invasive ventilation, research findings suggest that dysphagia is prevalent in the population of critically ill patients with COVID-19, with 55–93% of patients experiencing swallowing difficulties [80,81]. The added challenge of COVID-19 further complicates dysphagia management as strict infection control measures, increased number of patients requiring ICU admission and limiting practitioners’ time and resources to care for all [82] may hinder comprehensive swallowing assessments and rehabilitative interventions. As a result, healthcare providers need to prioritize the early identification and management of dysphagia in critically ill COVID-19 patients to mitigate the potential complications and improve patient outcomes.

Hopefully, mapping the current situation will add to the knowledge base required to produce international guidelines for PED management. Importantly, the severe clinical complications associated with PED in ICU patients [11,39,83] dictate that a comprehensive unit-based dysphagia education program must be urgently implemented to positively affect the uptake of therapeutic interventions and improve the quality of care provided. 

Looking to the future of care of ICU patients, artificial intelligence (AI) chatbots like ChatGPT (OpenAI, San Francisco, CA, USA) can contribute to the individualized medical care of critically ill patients with dysphagia. By analyzing patient data (e.g., medical records, imaging scans and vital signs), AI can play a significant role in early detection of dysphagia, timely recognition of changes that may indicate worsening dysphagia or related complications, and risk assessment of complications associated with dysphagia. In this way, healthcare professionals can intervene promptly, implement appropriate treatment plans and be alerted to take timely action. Additionally, by considering patient-specific factors like comorbidities, medication interactions and swallowing function assessments, AI can help develop individualized treatment plans for ICU patients with dysphagia. AI can also recommend tailored interventions, including dietary modifications and rehabilitation exercises. As patients with dysphagia have to overcome communication barriers, AI-powered speech recognition systems can assist them in communicating their needs, difficulties and discomforts more effectively, improving the overall care experience. Finally, AI can serve as a decision support tool for healthcare professionals by providing them with evidence-based recommendations, relevant research findings and treatment guidelines specific to dysphagia management. This can assist clinicians in making informed decisions and improving patient outcomes [84,85].

It is important to note that while AI can provide valuable assistance, it should always be used in conjunction with clinical expertise, skills and knowledge of healthcare professionals to deliver the best possible care to ICU patients with dysphagia. The interdisciplinary collaboration between nurses, intensivists and SLP/SLTs is a prerequisite for the success of any of these initiatives.

## Figures and Tables

**Figure 1 healthcare-11-02283-f001:**
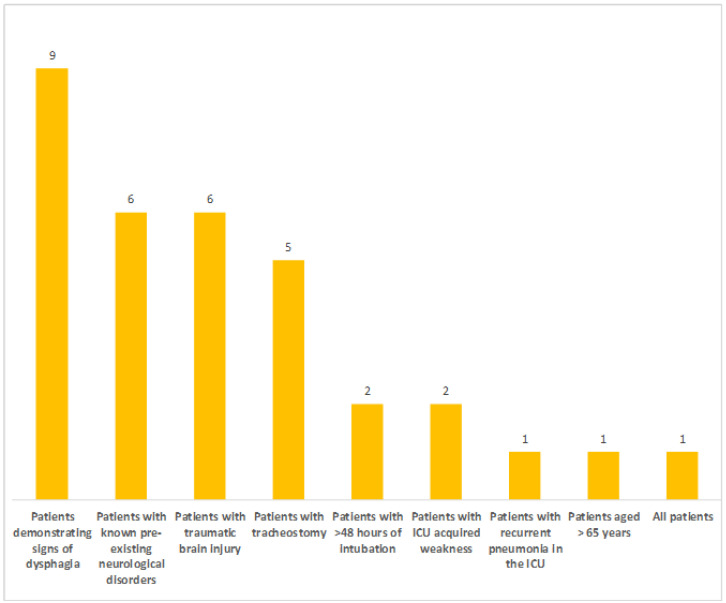
Patient screening for PED (ICUs were allowed to choose more than one answer) (numbers on the top of the bar graph refer to the number of ICUs).

**Figure 2 healthcare-11-02283-f002:**
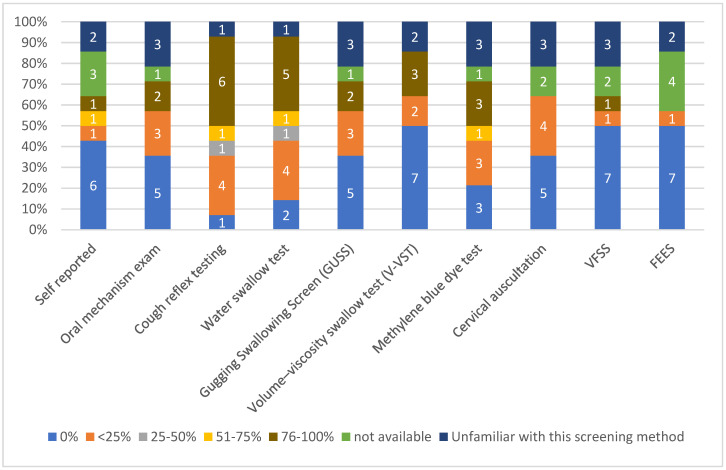
Assessment methods used to detect oropharyngeal dysphagia (numbers in the bar graph refer to the number of ICUs).

**Figure 3 healthcare-11-02283-f003:**
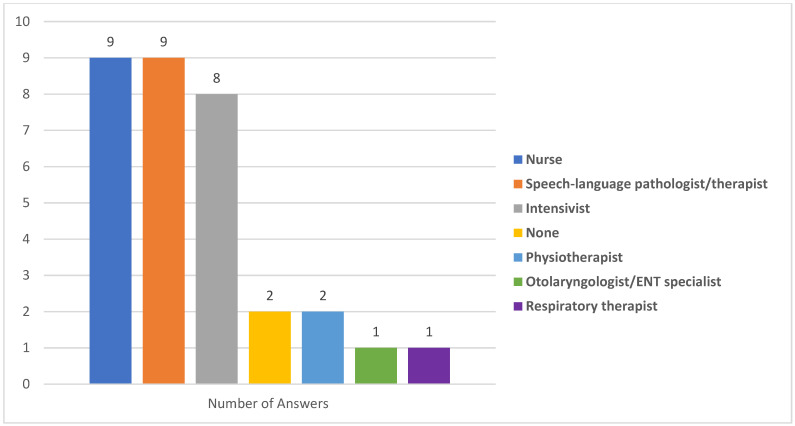
Team member who assesses for PED (numbers on the top of the bar graph refer to the healthcare personnel reported by ICUs as the team member who assesses patients for PED).

**Figure 4 healthcare-11-02283-f004:**
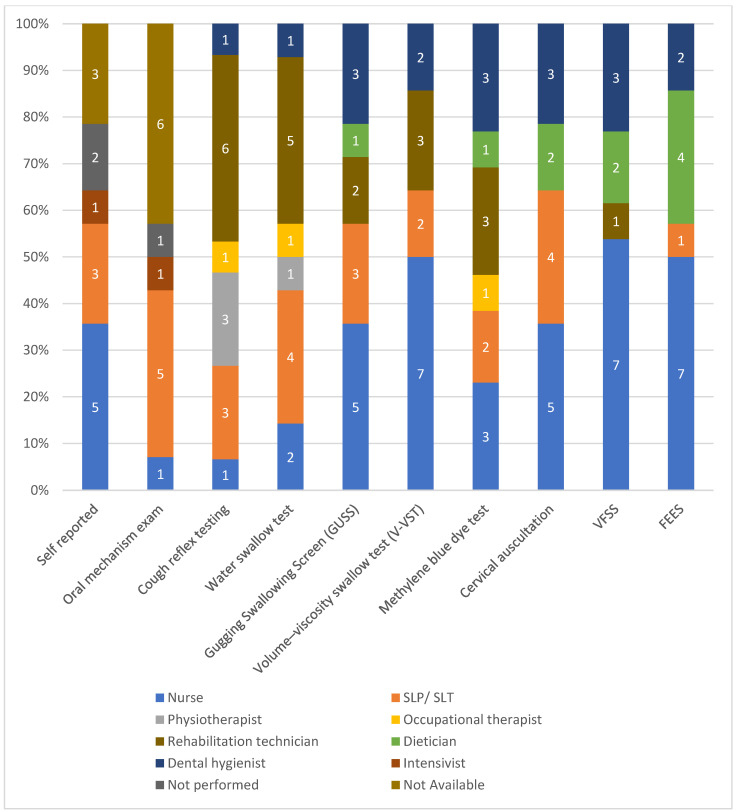
Responsibilities of each ICU member in screening methods used for PED (ICUs were allowed to choose more than one answer) (numbers on the bar graph refer to the ICU member who performs the tests).

**Figure 5 healthcare-11-02283-f005:**
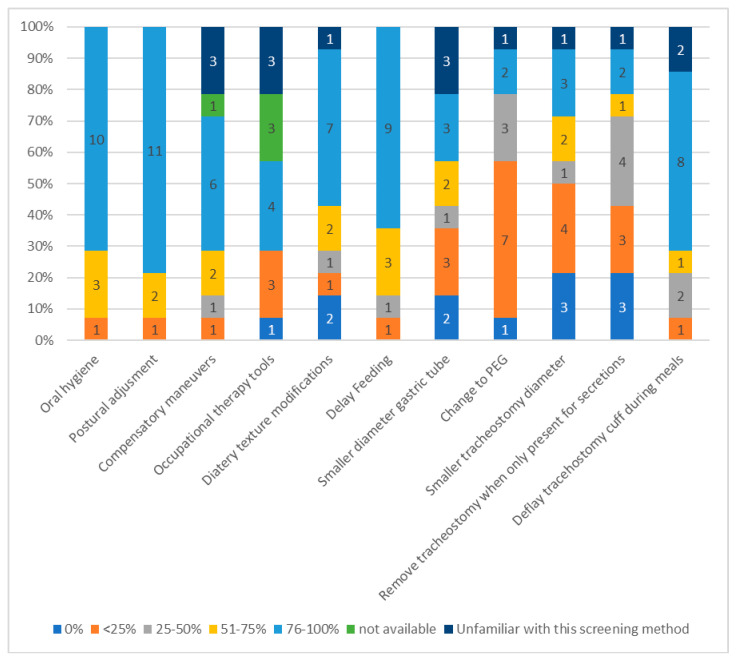
Measures taken to prevent aspiration/aspiration pneumonia related to PED (numbers in the bar graph refer to the chosen percentage per ICU).

**Figure 6 healthcare-11-02283-f006:**
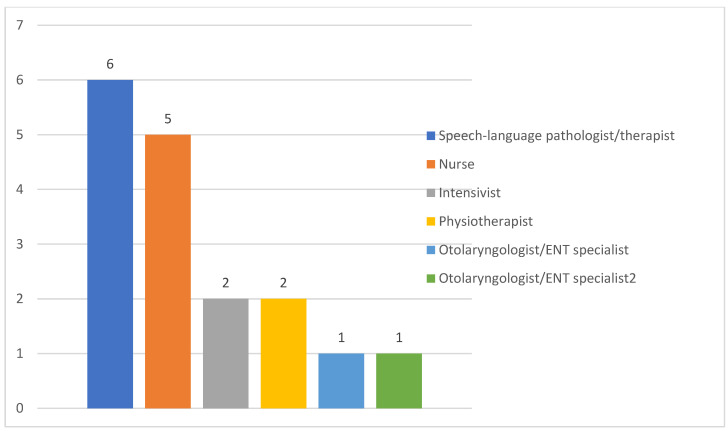
Team member who performs bedside swallow training.

**Table 1 healthcare-11-02283-t001:** ICU demographics.

	ICU Patient Capacity in Beds	ICU Type	Hospital Capacity	SLP/SLT Available
1	5–9 **	Mixed medical/surgical	<200	No
2	5–9 *	Mixed medical/surgical/neurosurgical	<200	No
3	5–9 *	Mixed medical/surgical	200–499	No
4	5–9 *	Mixed medical/surgical	<200	No
5	5–9 **	Medical/surgical/Neurosurgical/cardiothoracic	<200	Yes, not ICU-dedicated
6	5–9 **	Mixed medical/surgical	<200	No
7	5–9 **	Medical/surgical/Neurosurgical/cardiothoracic	<200	Yes, not ICU-dedicated
8	10–14 **	Mixed medical/surgical	<200	Yes, not ICU-dedicated
9	10–14 *	Mixed medical/surgical	200–499	No
10	10–14 *	Mixed medical/surgical	<200	Yes, not ICU-dedicated
11	15–19 *	Medical/surgical/Neurosurgical/cardiothoracic	200–499	Yes, not ICU-dedicated
12	10–14 *	Coronary Unit	200–499	Yes, not ICU-dedicated
13	15–19 *	Mixed medical/surgical	200–499	Yes, not ICU-dedicated
14	5–9 *	Burns Unit	200–499	Yes, not ICU-dedicated

* Public Hospital; ** Private hospital; SLP/SLT: speech and language pathologist/therapist.

**Table 2 healthcare-11-02283-t002:** Interventions to treat dysphagia.

Percentage of Patients							
	0%	<25%	25–50%	51–75%	>75%	Not Available	Unfamiliar with This Intervention
Intervention used to treat PED							
Repetitive swallowing exercises/maneuvers with or without additional resistance (e.g., Mendelsohn or Masako maneuver, supraglottic swallow)	3	1	3	1	1		5
Muscle-strengthening exercises without swallowing (e.g., chin tuck against resistance or Shaker exercise)	3	3	0	1	2		5
Muscle-strengthening exercises using apps on a tablet/iPad	7	1	0	0	1		5
Respiratory exercises [e.g., expiratory muscle strength training (EMST)]	5	0	0	2	3		4
Neuromuscular electrical stimulation (NEMS) of swallowing muscles	7	1	1	0	0	1	4
Surface EMG (sEMG) biofeedback swallowing training	8	1				1	4
Pharyngeal electrical stimulation (PES)	8					2	4

**Table 3 healthcare-11-02283-t003:** Awareness of PED consequences; IQR: interquartile range.

Survey Item	Proportion in Agreement	Mean (Standard Deviation)	Median (IQR)	Modal Value (Appearance Times)
Oropharyngeal dysphagia influences ICU length of stay	12/14 (85.7%)	5.64 (1.33)	6 (1)	6 (7)
Oropharyngeal dysphagia influences hospital length of stay	10/14 (71.4%)	4.57 (2.4)	6 (5)	6 (6)
Oropharyngeal dysphagia influences the delay in return to independent physical functioning after critical illness	14/14 (100%)	6.57 (0.64)	7 (1)	7 (9)
Oropharyngeal dysphagia influences the need for care at long-term facilities or nursing homes after critical illness	14/14 (100%)	6.35 (0.63)	6 (1)	6 (7)
The presence of oropharyngeal dysphagia influences the risk of ICU-readmission	10/14 (71.4%)	4.85 (2.03)	5 (5)	2, 5, 7 (4 each)

**Table 4 healthcare-11-02283-t004:** Screening methods that should be available in the ICU for PED.

Water Swallow Test (Including the Yale Swallow Protocol)
Gugging Swallowing Screen (GUSS)
Volume-viscosity swallow test
Oral mechanism exam
Methylene (Evan’s) blue dye test
Cervical auscultation
Video fluoroscopic swallowing study (VFSS)
Fiberoptic endoscopic evaluation of swallowing (FEES)

**Table 5 healthcare-11-02283-t005:** Methods that should be available in the ICU for the treatment of PED.

No Need for Dysphagia-Specific Treatment, the Dysphagia will Disappear when the Patient’s Strength Increases
Protocolized changing in fluid consistency and texture
Protocolized postural changes (chin down, etc.)
Repetitive swallowing exercises/maneuvers (e.g., Mendelsohn or Masako maneuver, supraglottic swallow)
Muscle-strengthening exercises without swallowing (e.g., chin tuck against resistance or Shaker exercise)
Muscle-strengthening exercises using apps on a tablet/iPad
Respiratory exercises (e.g., expiratory muscle strength training (EMST))
Smaller bore gastric feeding tube
Change to PEG-tube
If tracheostomy is present, replace with a smaller cannula tube
If tracheostomy is present only because of managing airway secretions, remove entirely
Neuromuscular electrical stimulation (NEMS) of swallowing muscles
Surface EMG biofeedback swallowing training
Pharyngeal electrical stimulation

## Data Availability

The datasets generated and/or analyzed during the present study are available from the corresponding author upon reasonable request.

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
