# Peer review of "Strategies of Screening and Treating Post-Extubation Dysphagia: An Overview of the Situation in Greek-Cypriot ICUs"

_healthcare, 2023, doi:10.3390/healthcare11162283_

Round 1

Reviewer 1 Report

The paper focuses on the screening of post-extubation dysphagia (PED) in Greek-Cypriot ICUs. It presents a secondary analysis of an international cross-sectional survey conducted in 26 countries. Overall, the paper is well-written and provides a local perspective on the original DICE study. The authors effectively highlight the issues related to PED in their country. However, the findings of this study do not significantly differ from those of the original study. The lack of awareness, screening, and protocols for PED is a common global issue. Consequently, I have concerns about the paper's contribution to the existing literature. 

Furthermore, I suggest reorganizing the reporting of data in a descending order, for "Figure 1, 3, 6." Similarly, in certain paragraphs such as "3.1.a. PED management".

Author Response

Reviewer 1

Reviewer’s Comment

Response

Citation (section and line/-s)

The paper focuses on the screening of post-extubation dysphagia (PED) in Greek-Cypriot ICUs. It presents a secondary analysis of an international cross-sectional survey conducted in 26 countries. Overall, the paper is well-written and provides a local perspective on the original DICE study. The authors effectively highlight the issues related to PED in their country.

We thank the reviewer for this comment.

However, the findings of this study do not significantly differ from those of the original study. The lack of awareness, screening, and protocols for PED is a common global issue.

Consequently, I have concerns about the paper's contribution to the existing literature.

We thank the reviewer for this comment which helped us to increase the rigor of our study.

We addressed this comment by adding the following text in the Limitation section:

First, this study presents a secondary analysis of an international cross-sectional survey conducted in 26 countries [27]. However, we still believe that our findings can provide a contribution to the existing literature as the context and environment of care can vary significantly among countries. Based on the above, regarding the results from Greek-Cypriot ICUs, this manuscript presented differences that are discussed in the discussion section compared to the findings of the DICE study [e.g The ICU teams in Cyprus seemed more aware of the risk of dysphagia to increase ICU and hospital stay (86% and 71% of respondents respectively) compared to the findings of the DICE study (64% and 42% respectively)]. Additionally, the results presented are more detailed. Fi-nally, as, to the best of our knowledge, there is a small number of nationwide studies about the management of post-extubation dysphagia and awareness of best practices in ICUs (e.g. Swiss survey of dysphagia care [43], the MADICU study in Germany [23] and a Dutch national ICU survey [36]), reporting results from individual countries is im-portant in informing quality improvement and education efforts. Not publishing these results would imply similar trends in a specific country, when in fact this is not the case.

Limitations, lines 599-613

Furthermore, I suggest reorganizing the reporting of data in a descending order, for "Figure 1, 3, 6."

Similarly, in certain paragraphs such as "3.1.a. PED management".

We thank the reviewer for this comment which helped us to increase the clarity of our study.

In figures 1, 3, 6 the data changed to descending order. 

This part of the Results section was reorganized in the way shown below (reporting of data in a descending order):

More than 85% (12 ICUs) of ICUs indicated that there was no standard protocol indicating which patients should be screened for PED.

9 ICU teams reported that no patient was screened for PED in their ICU. 2 ICUs reported that 51-75% of their patients screened for PED and in 1 ICU the percentage was 25-50% of patients. Screening for PED for less than 25% of their patients was reported in 1 ICU and only 1 ICU team reported that more than 75% of their patients screened for PED.

6 ICU teams reported that more than 75% of their patients who received a tracheostomy during their ICU admission screened for PED. 3 ICU teams reported that 51-75% of their tracheotomized patients screened for PED, in 3 ICUs the percentage was 25-50% of their patients and another 3 teams reported that no patient with tracheostomy was screened for PED in their ICU. Screening for PED after tracheostomy for less than 25% of patients was reported in 1 ICU.

Figures (pages 6,8,12)

Results

3.1.a. PED management (lines 208-219)

Reviewer 2 Report

Thanks for the opportunity to review this manuscript! Few points to further improve the paper are listed below:

Abstract: Overall, the abstract provides a good overview of the study and its key findings. However, it would benefit from providing more specific details, such as sample size, survey response percentages, and supporting statistics in the background section.

Introduction:

1. Clarity and Structure: The introduction provides a clear definition of oropharyngeal dysphagia (OD) and its classification in the International Classification of Diseases (ICD). However, the flow of the introduction could be improved by organizing the information more logically. Currently, there are abrupt transitions between discussing OD in general, its prevalence in various patient populations, dysphagia in critically ill patients, and the specific focus on post-extubation dysphagia (PED). It could be helpful to restructure the introduction to provide a smoother progression of information.

2. Literature Review: The introduction includes some relevant literature references to support the prevalence and impact of dysphagia. However, the introduction could benefit from incorporating more recent references to demonstrate up-to-date knowledge on the subject.

3. Gap Identification: The introduction successfully highlights the gap in routine bedside screening for PED and attributes it to limited awareness and inadequate knowledge among healthcare professionals. However, it does not provide specific evidence or examples to support this claim. Including specific studies or data on the lack of PED screening in ICUs would strengthen the argument and emphasize the need for the current study.

4. Research Objectives: The research objectives are stated clearly, indicating the aim of establishing current approaches to PED screening, management, and treatment in Greek Cypriot ICUs, as well as assessing ICU awareness of PED and its consequences. These objectives align with the purpose of the study.

5. Contextualization: The introduction briefly mentions the increase in life expectancy and the higher number of elderly patients being admitted to ICUs. While this information provides some context, it could be further developed by discussing the specific relevance of these factors to PED and the importance of addressing dysphagia in ICU patients.

6. Research Gap in Cyprus: The introduction briefly mentions that the prevalence of PED in Cyprus has not been identified and that there are no known protocols for assessing and treating dysphagic patients in ICUs. However, it does not explain why this is a significant research gap or how it contributes to the broader understanding of PED. Providing more context and justification for focusing on the situation in Greek-Cypriot ICUs would enhance the introduction.

Methods:

1. Study Design and Settings: The study design is clearly stated as a cross-sectional online survey conducted as part of the international survey "Dysphagia in Intensive Care Evaluation (DICE)" in which Cyprus participated. However, additional information about the specific timeframe of the survey and how it was administered would be helpful for better understanding the data collection process.

2. Participants: The participants are described as senior nurses from all adult Greek-Cypriot ICUs. The method of recruitment and the criteria for selecting senior nurses from each ICU are provided. However, it would be beneficial to clarify the total number of participating ICUs and the sample size in terms of the number of senior nurses who completed the questionnaire.

3. Data Collection and Instrument: The use of an anonymous self-administered survey is described, and the source of the questionnaire is referenced. The questionnaire's content is briefly outlined, including the types of questions and domains covered. However, there is no mention of any pilot testing or validation of the questionnaire for the specific context of Greek-Cypriot ICUs? Providing some information on the validity and reliability of the questionnaire would enhance the methodological rigor of the study.

4. Translation of the Instrument: The translation process is described, involving translation from English to Greek and back-translation by independent bilingual academics. The involvement of an external panel of experts in reviewing the translated questionnaire is mentioned. However, it would be valuable to provide more details about the specific steps taken during the translation process to ensure linguistic and cultural equivalence.

5. Ethics Approval: The study protocol's approval by the National Committee of Bioethics of Cyprus is mentioned, and submission of the completed questionnaire is considered as participation after informed consent. This demonstrates compliance with ethical guidelines. However, it may be helpful to clarify whether any confidentiality measures were implemented to ensure the anonymity of participants and data protection.

6. Data Analysis: The data collection method using Google Forms and the subsequent extraction and analysis of data in Excel are described. Descriptive statistics and non-parametric Spearman correlation are mentioned as the analytical methods used. However, there is no mention of any statistical tests or significance levels applied in the analysis. Providing more information on the specific statistical analyses conducted would strengthen the validity of the study findings.

Results:

1. Participants: The description of the participants and the characteristics of the participating ICUs is clear and provides relevant information, such as the number of beds and the types of patients admitted. However, it would be helpful to provide the total number of ICUs in the Republic of Cyprus to understand the representativeness of the sample. Additionally, the lack of dedicated SLP/SLTs in all ICUs and the availability of SLP/SLTs upon request should be further clarified as they could impact the management of PED.

2. Current Practices on PED:

   a. PED Management: The presentation of the data on existing protocols and subgroups of patients screened provides a clear overview of the current practices. However, it would be beneficial to discuss the implications of not having a standard protocol for PED screening and the potential impact on patient outcomes.

   b. Timing of Screening: The data on the timing of dysphagia screening are adequately presented. However, it would be valuable to discuss the significance of early screening for PED and the potential benefits of timely intervention.

   c. Methods Used for PED Assessment: The data on the methods used for PED assessment are well-presented and provide insights into the common practices in the ICUs. However, it would be beneficial to discuss the limitations of relying on certain assessment methods and the potential benefits of utilizing more comprehensive and instrumental assessments, such as VFSS and FEES.

3. Prevention of Aspiration Pneumonia Related to PED: The data on interventions to prevent aspiration pneumonia are clearly presented. However, it would be useful to provide a discussion on the effectiveness and evidence base for each intervention and the challenges in implementing them consistently across ICUs.

4. Interventions to Treat PED: The data on interventions to treat PED are adequately presented. However, it would be beneficial to discuss the limitations of relying on muscle-strengthening exercises without swallowing as the most common intervention and the potential benefits of a more comprehensive and targeted approach involving multidisciplinary collaboration.

5. Scope of the Problem: The data on ICU awareness of PED incidence and consequences are well-presented. However, it would be valuable to discuss the implications of the findings, such as the need for increased awareness and education among healthcare professionals and the potential impact on patient outcomes and healthcare resource utilization.

6. Perceived Best Practices on PED: The data on the perception of best practices are adequately presented. However, it would be beneficial to discuss the importance of standardized protocols and the barriers and facilitators identified by the ICUs in implementing these practices. Additionally, it would be useful to explore the potential implications of the identified barriers and facilitators in improving the quality of care for patients with PED.

Discussion:

Here are a few thoughts on the Discussion section:

1. Overview and Objectives: The introduction to the discussion provides a clear overview of the study's objectives and the purpose of the survey. However, it would be beneficial to explicitly state the main findings of the study to guide the subsequent discussion.

2. Perceived Best and Current Practices on PED: The discussion of the perceived best and current practices on PED is comprehensive and provides relevant insights. The comparison with findings from other studies adds value to the discussion. However, it would be helpful to further elaborate on the implications of the absence of standardized dysphagia assessment protocols and the limited availability of SLP/SLTs in ICUs. Discussing the potential consequences of these factors on patient outcomes and the need for interdisciplinary collaboration would enhance the discussion.

3. Methods Used for PED Assessment: The discussion of the methods used for PED assessment adequately highlights the limitations of relying on certain assessment methods, such as the water swallow test, and the potential benefits of instrumental assessments like VFSS and FEES. However, it would be beneficial to discuss the challenges and barriers to implementing these instrumental assessments in ICUs, such as the availability of technology and trained SLP/SLTs.

4. Responsibilities of ICU Team Members: The discussion of the responsibilities of ICU team members in PED assessment is informative. However, it would be valuable to further discuss the implications of relying on nurses and intensivists for PED assessment and the potential benefits of involving dedicated SLP/SLTs in the ICU team. Exploring the challenges and opportunities for expanding the role of nurses and implementing interdisciplinary collaboration could enrich the discussion.

5. Prevention and Treatment of PED: The discussion of the interventions to prevent and treat PED is concise and provides relevant information. However, it would be useful to discuss the limitations of the current interventions, such as muscle-strengthening exercises without swallowing, and the potential benefits of more comprehensive and targeted interventions. Addressing the challenges in implementing these interventions consistently across ICUs and the need for further research on effective treatment strategies would enhance the discussion.

6. Awareness of PED Incidence and Consequences: The discussion of ICU team awareness of PED incidence and consequences is well-presented. However, it would be beneficial to further discuss the potential reasons for the variation in awareness among ICU teams, such as differences in healthcare systems, interdisciplinary collaboration, and educational programs. Exploring strategies to improve awareness and knowledge among ICU teams could strengthen the discussion.

7. The Discussion section provides a comprehensive overview of the survey findings and highlights important aspects related to the awareness, practices, and challenges of managing post-extubation dysphagia (PED) in ICUs. However, it would be valuable to incorporate a paragraph discussing the potential implications of the results in the context of the COVID-19 pandemic. 

8. Any possible future roles of AI chatbots (like ChatGPT) in the future towards the individualized medical care of these ICU patients? (Examples: PMID: 37223340 and PMID: 37179277 )

9. Limitations: The discussion of the study's limitations is appropriate and acknowledges the potential biases and limitations of the survey. However, it would be helpful to further discuss the implications of these limitations on the generalizability and validity of the study findings, especially in the post-COVID era? Exploring the potential impact of the limitations on the interpretation of the results and suggesting directions for future research would add depth to the discussion.

The Conclusion section provides a concise summary of the study findings and emphasizes the need for improved awareness and knowledge regarding PED among ICU teams in Cyprus. It also highlights the importance of establishing protocols for the screening and treatment of dysphagia and emphasizes the severity of clinical complications associated with PED in ICU patients.

There are no significant English language issues. The manuscript is coherent, well-structured, and effectively conveys the main points of the research.

Author Response

Reviewer 2

Reviewer’s Comment

Response

Citation (section and line/-s)

Thanks for the opportunity to review this manuscript! Few points to further improve the paper are listed below:

We thank the reviewer for his comments which helped us to increase the rigor of our study.

Abstract: Overall, the abstract provides a good overview of the study and its key findings.

However, it would benefit from providing more specific details, such as sample size, survey response percentages, and supporting statistics in the background section.

We thank the reviewer for this comment.

We made the suggested additions to the abstract section (underlined texts):

Post-extubation dysphagia (PED) can lead to serious health problems in critically ill patients. Contrasting its high incidence rate of 12.4% reported in a recent observational study, many ICUs lack routine bedside screening, likely due to limited awareness. This study aimed to establish baseline data on the current approaches, the status of perceived best practices to PED screening and treatment, as well as to assess awareness of PED. A nationwide cross-sectional, online survey was conducted in all fourteen adult ICUs in the Republic of Cyprus in June 2018, with a 100% response rate.

Abstract (lines 14-19)

Introduction:

1. Clarity and Structure: The introduction provides a clear definition of oropharyngeal dysphagia (OD) and its classification in the International Classification of Diseases (ICD).

However, the flow of the introduction could be improved by organizing the information more logically. Currently, there are abrupt transitions between discussing OD in general, its prevalence in various patient populations, dysphagia in critically ill patients, and the specific focus on post-extubation dysphagia (PED). It could be helpful to restructure the introduction to provide a smoother progression of information.

We thank the reviewer for this comment which helped us to organize the information more logically.

This part of Introduction section was restructured in the way shown below (underlined text):

Oropharyngeal dysphagia (OD) is defined as the difficulty in transferring liquids and food from the oropharyngeal cavity into the stomach, and it refers to any abnormality in the swallowing physiology of the upper aerodigestive tract [1). It is listed under code MD93 in the International Classification of Diseases (ICD), 11th revision [2]. In a recent systematic review and meta-analysis the global prevalence of OD was 43.8% in different populations while a sub-group analysis showed that it was high (48.1%) in elderly population [3]. Researchers demonstrated that the prevalence of diagnosed OD was 3% among adult patients admitted to hospitals in the United States [4] and, in consistency with the previous results, it was higher in the population over 75 years old than other age groups [5]. OD is a common comorbidity among various patient populations as stroke and cancer patients and patients with brain injury [6–8]. Critically ill patients are also a population in high-risk to develop dysphagia as it is a significant healthcare-acquired complication due to invasive mechanical ventilation.

Introduction (lines 35-44)

2. Literature Review: The introduction includes some relevant literature references to support the prevalence and impact of dysphagia.

However, the introduction could benefit from incorporating more recent references to demonstrate up-to-date knowledge on the subject.

We thank the reviewer for this comment which helped us to demonstrate up-to-date knowledge on the subject.

We replaced the following references with more recent ones:

Previous ref:

Okuni, I.; Otsubo, Y.; Ebihara, S. Molecular and neural mechanism of dysphagia due to cancer. Int J Mol Sci 2021, 22, 7033. doi: 10.3390/ijms22137033

New ref:

Tanaka A, Uemura H, Kimura T, Nishimura A, Aoki K, Otsuka S, Ueda K, Kitahara T. Evaluation of usefulness of tongue pressure measurement device for dysphagia associated with treatment of patients with head and neck cancer (ELEVATE). Medicine (Baltimore). 2023 Jun 30;102(26):e33954. doi: 10.1097/MD.0000000000033954.

Previous ref:

Cabré, M.; Almirall, J.; Clavé, P. Aspiration pneumonia: management in Spain. Eur Geriatr Med 2011, 2, 180–183. doi: 10.1016/J.EURGER.2011.03.004

New ref:

Eskildsen SJ, Jakobsen D, Riberholt CG, Poulsen I, Curtis DJ. Protocol for a scoping review study to identify and map treatments for dysphagia following moderate to severe acquired brain injury. BMJ Open. 2019 Jul 17;9(7):e029061. doi: 10.1136/bmjopen-2019-029061.

Previous ref:

Skoretz, S.A.; Flowers, H.L.; Martino, R. The incidence of dysphagia following endotracheal intubation: a systematic review. Chest 2010, 137, 665–673. doi: 10.1378/CHEST.09-1823

New ref:

Mc Intyre M, Doeltgen S, Dalton N, Koppa M, Chimunda T. Post Extubation dysphagia incidence in critically ill patients: a systematic review and meta-analysis. Aust Crit Care 2020. https://doi.org/10.1016/j.aucc.2020.

05.008.

Previous ref:

Medeiros, G.C.; Sassi, F.C.; Zambom, L.S.; de Andrade, C.R.F. Correlation between the severity of critically ill patients and clinical predictors of bronchial aspiration. J Bras Pneumol 2016, 42, 114–120. doi: 10.1590/S1806-37562015000000192

New ref:

Rofes L, Muriana D, Palomeras E, Vilardell N, Palomera E, Alvarez-Berdugo D, Casado V, Clavé P. Prevalence, risk factors and complications of oropharyngeal dysphagia in stroke patients: A cohort study. Neurogastroenterol Motil. 2018 Mar 23:e13338. doi: 10.1111/nmo.13338.

Previous ref:

Christensen, M.; Trapl, M. Development of a modified swallowing screening tool to manage post-extubation dysphagia. Nurs Crit Care 2018, 23, 102–107. doi: 10.1111/nicc.12333

New ref:

Viñas, P.; Martín-Martínez, A.; Alarcón, C.; Riera, SA,; Miró, J.; Amadó, C,; Clavé, P.; Ortega, O. A Comparative Study be-tween the Three Waves of the Pandemic on the Prevalence of Oropharyngeal Dysphagia and Malnutrition among Hospitalized Patients with COVID-19. Nutrients 2022, 16, 14, 3826. doi: 10.3390/nu14183826

Previous ref:

Macht, M.; Wimbish, T.; Clark, B.J.; Benson, A.B.; Burnham, E.L.; Williams, A.; Moss, M. Postextubation dysphagia is per-sistent and associated with poor outcomes in survivors of critical illness. Crit Care 2011, 15, R231 doi: 10.1186/cc10472

New ref:

Schefold, J.C.; Berger, D.; Zürcher, P.; Lensch, M.; Perren, A.; Jakob, S.M.; Parviainen, I.; Takala, J. Dysphagia in mechanically ventilated ICU patients (Dynamics): A prospective observational trial. Crit Care Med 2017, 45, 2061–2069. doi: 10.1097/CCM.0000000000002765

Previous ref:

Brodsky, M.B.; González-Fernández, M.; Mendez-Tellez, P.A.; Shanholtz, C.; Palmer, J.B.; Needham, D.M. Factors associated with swallowing assessment after oral endotracheal intubation and mechanical ventilation for acute lung injury. Ann Am Thorac Soc 2014, 11, 1545–1552. doi: 10.1513/AnnalsATS.201406-274OC

New ref:

Royals WJ, Gillis RJ, Campbell JL. A Decision Guide for Assessing the Recently Extubated Patient's Readiness for Safe Oral Intake. Crit Care Nurse. 2023 Feb 1;43(1):42-51. doi: 10.4037/ccn2023722.

References

3. Gap Identification: The introduction successfully highlights the gap in routine bedside screening for PED and attributes it to limited awareness and inadequate knowledge among healthcare professionals.

However, it does not provide specific evidence or examples to support this claim. Including specific studies or data on the lack of PED screening in ICUs would strengthen the argument and emphasize the need for the current study.

We thank the reviewer for this comment which helped us to emphasize the need for the current study.

We addressed this comment by adding the following, underlined text:

Limited awareness, and inadequate knowledge of healthcare professionals [19], especially of nurses [20-22] but also of physicians [23], are some of the reasons that were attributed to limited screening for PED in the intensive care units (ICUs).

We also added these references concerning nurses’ limited awareness and inadequate knowledge:

Christensen M, Trapl M. Development of a modified swallowing

screening tool to manage post-extubation dysphagia. Nurs Crit Care

2018;23:102e7.

Dallal York J, Miller S, Chapin J, Gore S, Jeng EI, Plowman EK. Swallowing screening practice patterns for nurses in the cardiac surgery intensive care unit. J Clin Nurs 2020;29:4573e82.

Nielsen AH, Kaldan G, Nielsen BH, Kristensen GJ, Shiv L, Egerod I. Intensive care professionals' perspectives on dysphagia management: A focus group study. Aust Crit Care. 2023 Jul;36(4):528-535. doi: 10.1016/j.aucc.2022.04.004.

We also added these references concerning physicians’ limited awareness and inadequate knowledge:

Marian, T., Dünser, M., Citerio, G. et al. Are intensive care physicians aware of dysphagia? The MADICU survey results. Intensive Care Med 44, 973–975 (2018). https://doi.org/10.1007/s00134-018-5181-1

Introduction (lines 62-63)

References

Lines 770-775

References

Lines 776-777

4. Research Objectives: The research objectives are stated clearly, indicating the aim of establishing current approaches to PED screening, management, and treatment in Greek Cypriot ICUs, as well as assessing ICU awareness of PED and its consequences. These objectives align with the purpose of the study.

We thank the reviewer for this comment.

5. Contextualization: The introduction briefly mentions the increase in life expectancy and the higher number of elderly patients being admitted to ICUs.

While this information provides some context, it could be further developed by discussing the specific relevance of these factors to PED and the importance of addressing dysphagia in ICU patients.

We thank the reviewer for this comment which helped us to increase the rigor of our study.

This part of Introduction section was restructured in the way shown below:

Recent screening studies in ICUs reported that patients with frailty represents approximately 30% of critically ill patients [28,29]. Frailty, among other things, is associated with prolonged ICU stay and mechanical ventilation [28-30]. Furthermore, as there has been an increase in life expectancy globally, a higher number of elderly patients will be admitted to the ICUs. Given that dysphagia has been associated with mechanical ventilation, age, and frailty [31,32] it is apparent that dysphagia is a critical area of concern in ICU patients. But these data will only bring value if they contribute to the way we provide care. Based on this, PED awareness is an important factor in screening, early diagnosis and treatment of dysphagia, as early identification is positively associated with treatment interventions [33].

Introduction (lines 69-78)

6. Research Gap in Cyprus: The introduction briefly mentions that the prevalence of PED in Cyprus has not been identified and that there are no known protocols for assessing and treating dysphagic patients in ICUs.

However, it does not explain why this is a significant research gap or how it contributes to the broader understanding of PED. Providing more context and justification for focusing on the situation in Greek-Cypriot ICUs would enhance the introduction.

We thank the reviewer for his comment which helped us to enhance the Introduction section.

We addressed this comment by adding the following text:

Within the specific context of Cyprus, there exists a significant research gap regarding the prevalence of PED. The absence of prior identification and understanding of PED in Cypriot ICUs represents a substantial knowledge deficiency that needs to be addressed. By investigating the prevalence of PED in Greek-Cypriot ICUs we can con-tribute to a more comprehensive understanding of dysphagia in the ICU setting and underscore the importance of investigating and addressing PED prevalence in Greek-Cypriot ICUs.

Introduction (lines 79-84)

Methods:

1. Study Design and Settings: The study design is clearly stated as a cross-sectional online survey conducted as part of the international survey "Dysphagia in Intensive Care Evaluation (DICE)" in which Cyprus participated.

However, additional information about the specific timeframe of the survey and how it was administered would be helpful for better understanding the data collection process.

We thank the reviewer for his comment.

This information was on the Translation and cultural adaptation of the instrument section. It was our mistake.

We removed the text A Google Forms link to the survey was emailed by the principal investigator to the participating ICUs senior nurses in June 2018. Only two email reminders were sent to non-responder nurses of four ICUs at a one-week intervals from the Translation of the instrument section and we added it to the Study design and settings section.

Design and settings (lines 96-98)

2. Participants: The participants are described as senior nurses from all adult Greek-Cypriot ICUs. The method of recruitment and the criteria for selecting senior nurses from each ICU are provided.

However, it would be beneficial to clarify the total number of participating ICUs and the sample size in terms of the number of senior nurses who completed the questionnaire.

We thank the reviewer for this comment.

We addressed this comment by adding the following text:

…the participants were all the fourteen adult ICUs in the Republic of Cyprus designated for the management of critically ill, intubated patients. This was the final sample size of the study as there were no exclusion criteria.

We also added ‘14’ in the Design and settings section

Participants (lines 102-105)

Design and settings (line 95)

3. Data Collection and Instrument: The use of an anonymous self-administered survey is described, and the source of the questionnaire is referenced. The questionnaire's content is briefly outlined, including the types of questions and domains covered.

However, there is no mention of any pilot testing or validation of the questionnaire for the specific context of Greek-Cypriot ICUs? Providing some information on the validity and reliability of the questionnaire would enhance the methodological rigor of the study.

We thank the reviewer for this comment which helped us to increase the methodological rigor of our study.

We addressed this comment by adding the following texts to the Translation and cultural adaptation of the instrument section:

To assess the face validity of the translated questionnaire, it was reviewed by an external panel of experts consisting of two academics with ICU experience for more than 10 years and four post-graduate nursing students: two Ph.D. students and two master’s students, all familiar with the Cypriot ICU context. None of the reviewers reported anything that was ambiguous or hard to com-prehend. The Cronbach’s alpha for the translated questionnaire was 0.96.

We also added the following text to the Limitation section:

Finally, although we checked internal reliability and face validity of the questionnaire used, several other forms of reliability and validity were not performed, e.g. test-retest analysis, concurrent validity, etc. This limitation suggests that the robustness and accuracy of the study's findings may be compromised. Future research should in-clude test-retest analysis to evaluate the stability of the questionnaire over time and concurrent validity testing to establish its relationship with other relevant measures.

Translation and cultural adaptation of the instrument (lines 157-162)

Limitations (lines 637-642)

4. Translation of the Instrument: The translation process is described, involving translation from English to Greek and back-translation by independent bilingual academics. The involvement of an external panel of experts in reviewing the translated questionnaire is mentioned.

However, it would be valuable to provide more details about the specific steps taken during the translation process to ensure linguistic and cultural equivalence.

We thank the reviewer for this comment which helped us to increase the methodological rigor of our study as ensuring linguistic and cultural equivalence during a translation process is essential to maintain the accuracy and intended meaning of the original text.

We addressed this comment by adding the following text:

In this study, we adopted the Report of the International Society for Pharmaco-economics and Outcomes Research (ISPOR) Task Force for Translation and Cultural Adaptation [35]. First, the questionnaire was translated from English to Greek by two independent Greek, bilingual academics, one specialized in evidenced-based practice and the other in critical care nursing. The Cypriot national coordinator of the study collected and combined the two versions, and a third Greek translation of the questionnaire was obtained. Then, another independent bilingual academic specialized in teaching and learning methodology in nursing translated the Greek version back into English without having read the original English version of the questionnaire. Finally, the Cypriot national coordinator of the study compared the original English version with the back-translated English version.

Translation and cultural adaptation of the instrument (lines 147-157)

5. Ethics Approval: The study protocol's approval by the National Committee of Bioethics of Cyprus is mentioned, and submission of the completed questionnaire is considered as participation after informed consent. This demonstrates compliance with ethical guidelines.

However, it may be helpful to clarify whether any confidentiality measures were implemented to ensure the anonymity of participants and data protection.

We thank the reviewer for this comment which helped us to clarify an important ethical issue.

We addressed this comment by adding the following texts:

Anonymity was assured as neither the identity of the senior nurse who completed the questionnaire nor the identity of any person of the ICU team were collected at any point of the study. Additionally, on the last line of the study’s cover letter was written that all data will remain deidentified, be only accessible to the research team and will be securely stored in password protected files.

Ethical approval (lines 166-170)

6. Data Analysis: The data collection method using Google Forms and the subsequent extraction and analysis of data in Excel are described. Descriptive statistics and non-parametric Spearman correlation are mentioned as the analytical methods used.

However, there is no mention of any statistical tests or significance levels applied in the analysis. Providing more information on the specific statistical analyses conducted would strengthen the validity of the study findings.

We thank the reviewer for this comment which helped us to strengthen the validity of the study findings. Nevertheless, the design and structure of the questionnaire, coupled with the limited surveyed population, restricted the scope of analysis to descriptive statistics only. Consequently, our focus was on summarizing and describing the collected data.

We addressed this comment by adding the following text:

Based on the design and structure of the questionnaire along with the limited surveyed population, our analysis was restricted to descriptive statistics only.

Data analysis (lines 182-184)

Results:

We thank the reviewer you for his valuable comments. Τhe Results section remains the same as in the first submission (presenting statistics without interpretation). The Discussion section’s headings have been rearranged to correspond to those in the Results section. All the comments the reviewer made for the Results section are addressed in the revised Discussion section.

1. Participants: The description of the participants and the characteristics of the participating ICUs is clear and provides relevant information, such as the number of beds and the types of patients admitted.

However, it would be helpful to provide the total number of ICUs in the Republic of Cyprus to understand the representativeness of the sample.

Additionally, the lack of dedicated SLP/SLTs in all ICUs and the availability of SLP/SLTs upon request should be further clarified as they could impact the management of PED.

We added the word Fourteen in the Results section.

We also added the following text in the Discussion section:

We conducted a survey involving health care teams working in all ICUs (14/14)

We addressed this comment by adding the following texts in the Discussion section:

Yet, to address the limited availability of SLP/SLTs in ICUs in Cyprus organizational choices are required at the level of health policy.

It would be very interesting to investigate how many SLPs/SLTs are available at the moment in Cyprus for ICU consultations including PED management.

What complicates the situation is that, according to the American Speech-Language-Hearing Association (ASHA), SLPs are the most qualified providers for dysphagia services and “cross-training of clinical skills is not appropriate at the professional level of practice" [46].

Until a dedicated SLP/SLT for PED screening and treatment becomes available for all ICUs, the empowerment of nurses through education along with the implementation of standardized protocols can contribute to the early identification of high-risk individuals for dysphagia and lead to referrals for optimal management. Apparently, professional and regulatory bodies of different health care professionals need to instigate interdisciplinary collaboration early in the education of undergraduate students which will hopefully lead to collaboration during clinical practice having patients as the point of reference.

The observed variation in the estimates of PED incidence and consequences be-tween the ICU teams in Cyprus might reflect differences between the institutions in terms of patients’ illness category and severity. However, it can also be an indication of the absence of available SLPs/SLTs for specialized consultation in ICUs in Cyprus.

What was seen as a barrier for standardized PED screening and treatment (lack of protocols and education) was reported as a facilitator if addressed at an organizational level. Interestingly, the collaboration with specially trained personnel was only reported as a facilitator possibly implying the complete unavailability of SLPs/SLTs and the anticipation by ICU teams to improve the care they offer to patients in the context of multidisciplinary collaboration. 

Results (line 193)

Discussion (line 371)

4.1.a.1 Existing protocols and subgroups of patients screened (lines 401-402)

4.1.a.4 Responsibilities of ICU team members (lines 444-445)

4.1.a.4 Responsibilities of ICU team members (lines 449-452)

4.1.a.4 Responsibilities of ICU team members (lines 457-464)

4.2.b Awareness of PED consequences (lines 556-560)

4.3.d Facilitators to standardized screening and treatment (lines 588-593)

2. Current Practices on PED:

a. PED Management: The presentation of the data on existing protocols and subgroups of patients screened provides a clear overview of the current practices.

However, it would be beneficial to discuss the implications of not having a standard protocol for PED screening and the potential impact on patient outcomes.

We addressed this comment by adding the following text in the Discussion section:

Most of the ICU teams in Cyprus reported the absence of any standardized dysphagia assessment protocol. The absence of an assessment protocol and screening procedures in the ICU is commonly reported by most investigators [23,36-38]. The percentage of ICU teams in Cyprus screening for dysphagia after tracheostomy is slightly improved and similar to practices in other countries [36,37] probably due to the perceived vulnerability of this population for dysphagia. The implications of not having a standard protocol for PED screening and the potential impact on patient outcomes [10, 11] point to an urgent need for the development of international guidelines for the screening and management of PED dysphagia. Based on the guidelines, new educational programs can be designed and implemented across countries to assure safe clinical practice. At the same time, the guidelines need to stress the necessity for interdisciplinary collaboration between ICU staff and SLP/SLTs due to the high level of specialization required for the management of PED and the unsettling consequences for patients if left untreated [39].

4.1.a.1 Existing protocols and subgroups of patients screened (lines 388-401)

b. Timing of Screening: The data on the timing of dysphagia screening are adequately presented.

However, it would be valuable to discuss the significance of early screening for PED and the potential benefits of timely intervention.

We addressed this comment by adding following texts in the Discussion section:

4.1.a.2 Timing of screening

After ICU admission

The practice of screening for dysphagia in the early stages of ICU admission is rarely practiced in Cyprus [no screening was reported by 9 ICU teams (64%)]. Even when it does take place, there is variation in the timing ranging from 24-48 hours (in 2 ICUs) to 3 to 7 days after admission (in 1 ICU). These findings are consistent with the lack of screening at admission reported in the literature [40] and probably reflects the lack of appreciation for the varying risk for dysphagia in different ICU patient groups.

After extubation

The same percentage of ICU teams reported that PED screening was performed at the day of extubation (28.5%) as well as 3 to 7 days after extubation (28.5%). No screening took place in 3 ICUs (21.4%). However, data from eight ICUs in Japan [41] found significant associations between each day of post-extubation delay in SLT con-sultation and dysphagia, aspiration pneumonia or death at the 7th, 14th, or 28th day after extubation. As a result, it is critical to appreciate timely post-extubation evaluations by trained professionals in order to implement timely interventions and prevent serious complications in high-risk ICU patients.

4.1.a.2 Timing of screening (lines 404-421)

c. Methods Used for PED Assessment: The data on the methods used for PED assessment are well-presented and provide insights into the common practices in the ICUs.

However, it would be beneficial to discuss the limitations of relying on certain assessment methods and the potential benefits of utilizing more comprehensive and instrumental assessments, such as VFSS and FEES.

We addressed this comment by adding the following text in the Discussion section:

Nevertheless, comprehensive and instrumental assessments, such as VFSS and FEES, are necessary for patients in the presence of clinical signs of aspiration when the water swallow screening is negative [38]. The observed paucity in their clinical application is attributed to the need for trained professionals and the availability of technological equipment which are mostly available at university hospitals [27].

4.1.a.3 Methods used for PED assessment (lines 431-435)

3. Prevention of Aspiration Pneumonia Related to PED: The data on interventions to prevent aspiration pneumonia are clearly presented.

However, it would be useful to provide a discussion on the effectiveness and evidence base for each intervention and the challenges in implementing them consistently across ICUs.

We addressed this comment by adding the following texts in the Discussion section:

4.1.b. Prevention of aspiration pneumonia related to PED

Postural adjustment, as well as oral hygiene, were the most widely used methods to decrease the risk of aspiration after suspected or confirmed PED in our study.

4.1.b.1 Aspiration/Aspiration pneumonia resulting from liquids/ solid food

  Postural adjustment has been proven to promote swallowing in patients with confirmed or suspected dysphagia by affecting bolus flow and speed, especially when the patient has been placed in a sitting position [50]. Importantly, irrespective of the bolus volume, manipulating the cervical and shoulder angle has been shown to activate more effectively swallowing-related muscles during thoracic upright sitting [51]. 

It is established that the maintenance of good oral hygiene decreases the risk of aspiration pneumonia in the ICU [52,53]. Although dysphagia is a recognized as a risk factor for aspiration pneumonia it is speculated that it contributes to its causation in combination with other risk factors such as poor oral hygiene [54]. As such, systematic oral hygiene can address the bacterial colonization of the oropharyngeal cavity and decrease the risk for dysphagia [55-57]. Furthermore, there is evidence that the cough reflex is improved with regular oral hygiene which can act synergistically to the reduc-tion of aspiration risk [58].

4.1.b.2 Aspiration pneumonia resulting from saliva production

Hypersalivation poses a serious aspiration risk for individuals with dysphagia since the normal clearance of secretions is impaired. The restricted use of saliva man-agement interventions by the ICU teams in our study probably depicts the lack of evi-dence in the published literature specifically for critically ill patients [59]. Similarly, hypersalivation due to swallowing difficulties have diverse aetiologies and multidisci-plinary collaboration is required to identify the causes and implement appropriate treatment.

The major challenge in implementing interventions for the prevention of aspiration pneumonia related to PED consistently across ICUs in the lack of studies on the topic specifically for critically ill patients. Focused research could try and replicate proven interventions from other populations in critical care and/or explore new both pharma-cological and non-pharmacological management options.

4.1.b. Prevention of aspiration pneumonia related to PED (lines 466-497)

4. Interventions to Treat PED: The data on interventions to treat PED are adequately presented.

However, it would be beneficial to discuss the limitations of relying on muscle-strengthening exercises without swallowing as the most common intervention and the potential benefits of a more comprehensive and targeted approach involving multidisciplinary collaboration.

We addressed this comment by adding the following text in the Discussion section:

4.1.c Interventions to treat PED

Muscle strengthening exercises without swallowing, repetitive swallowing exer-cises/maneuvers with or without additional resistance have been identified as the most widely used interventions to treat PED in our survey. Still, they were only used by a limited number of participating ICUs. Although there is proof that these exercises promote muscle strengthening [60-62] and consequently swallowing, recent advances in post-extubation therapy employ swallowing techniques aided by surface electromyog-raphy [63] as well as electrostimulation of the pharynx for dysphagia treatment [64,65] with promising results. The ICU teams in our study were not familiar with these new treatment modalities finding that needs further consideration. Certainly, introducing these approaches require a comprehensive approach involving targeted education and multidisciplinary collaboration which were identified mostly as unavailable in the ICUs of Cyprus.

4.1.c Interventions to treat PED (lines 500-511)

5. Scope of the Problem: The data on ICU awareness of PED incidence and consequences are well-presented.

However, it would be valuable to discuss the implications of the findings, such as the need for increased awareness and education among healthcare professionals and the potential impact on patient outcomes and healthcare resource utilization.

We addressed this comment by adding the following text in the Discussion section:

4.2 Scope of the problem

  4.2.a Awareness of PED incidence

Only 28.5% of the participating ICU teams recognized PED as common amongst ICU patients which suggests low awareness of dysphagia in the participating ICU teams. Our results were very similar to the findings reported by the Swiss survey of dysphagia care [43], the MAD-ICU study in Germany [23], and a Dutch national ICU survey [36] but lower than the frequency of OD occurrence reported in the DICE in-ternational study (47%) [27]. Many respondents in our survey thought that the duration of intubation and the presence of tracheostomy increase the PED occurrence which has also been demonstrated in the literature [9,66,67]. In 42.8% of the ICU teams the inci-dence of PED was estimated as 25-50% for patients who remained intubated for more than 7 days while in 21.4% of ICU teams between 51 to 75% which was less than the estimation of the DICE study (67%) [27]. The incidence of PED in patients with a tra-cheostomy was estimated to be 51–75% by most respondents in our study with 25–50% being the 2nd most frequent estimate. This is in accordance with the Dutch study [36] with cohorts including non-neurologic critically ill patients with a tracheostomy [68] as well as neurologic patients [69,70].

  4.2.b Awareness of PED consequences

The vast majority of the participating ICUs in our study agreed that the duration of ICU stay was associated with increased PED occurrence. Yet, the reasons for the pro-longed ICU stay were not known since no scoring system was used in the current study. However, there is evidence that ICU patients' disease severity in the Republic of Cyprus is high [52]. As patients with increased disease severity stay longer in the ICU and have a longer duration of intubation than others with less severe conditions [71,72], thus, they are more likely to develop PED [72]. Additionally, it is well evidenced that PED patients have a significantly longer LOS in hospitals in comparison to patients with normal swallowing [73–75], a finding that more than two-thirds of the participating ICUs in our study agreed with. The ICU teams in Cyprus seemed more aware of the contribution of dysphagia in the prolongation of ICU and hospital LOS compared to the findings of the DICE study [27]. Yet, it remains unclear whether PED resulted in the increased LOS or, whether the increased LOS resulted in PED in our participants' reports. A prospective observational study could help answer this question.

The functional status of a person is defined as the ability to carry out daily living abilities including eating [76]. Since PED has the potential to compromise eating ability in hospitalized patients, it can delay the return of a patient to pre-hospitalization functioning status, a finding that was also reported by all the participating ICUs of this study. Additionally, all participants agreed that patients who present with PED may need long-term care in comparison with patients without. Furthermore, in our study, almost all of the participating ICUs perceived that ICU readmission was associated with PED. Although, the association seems very possible it has not been positively correlated among patients with clinically significant PED compared to those without [77].

The observed variation in the estimates of PED incidence and consequences be-tween the ICU teams in Cyprus might reflect differences between the institutions in terms of patients’ illness category and severity. However, it can also be an indication of the absence of available SLPs/SLTs for specialized consultation in ICUs in Cyprus.

4.2 Scope of the problem (lines 513-560)

6. Perceived Best Practices on PED: The data on the perception of best practices are adequately presented.

However, it would be beneficial to discuss the importance of standardized protocols

and the barriers and facilitators identified by the ICUs in implementing these practices. Additionally, it would be useful to explore the potential implications of the identified barriers and facilitators in improving the quality of care for patients with PED.

We addressed this comment by adding the following text in the Discussion section:

4.3.a Protocols and routine screening

A large percentage of participating ICUs agreed that a standard protocol should be used for PED screening for patients who remained in the ICU and/or remained intubated longer than 48hours. In combination with the absence of protocols for PED screening and management currently in Cyprus is encouraging for future efforts to-wards evidence-based practices. It might signify that ICU teams are ready to welcome guidelines as soon as they are developed and implement them diligently at a national level.

4.3.b. Availability of screening and treating methods

Similarly, the fact that ICU teams in our study favored all PED screening and treating methods potentially attests to the level of readiness of ICU teams to instigate new practices.

4.3.c Barriers to standardized screening and treatment

The ICU teams of Cyprus seem to be conscious of the lack of evidence-based protocols for PED screening and treatment since they recognize it as the most important barrier in the implementation of standardized PED management. Evidence-based practice is strongly connected to the improvement of patient outcomes [78] and the importance of protocols for PED has already been emphasized. What merits further exploration is the association between the instigation of new protocols in naïve environments and the improvement of patient reported outcomes [79]. Additionally, educational institutions along with professional bodies need to employ research results to design courses that could address the educational gap identified by ICU teams for PED screening and treatment.

4.3.d Facilitators to standardized screening and treatment

What was seen as a barrier for standardized PED screening and treatment (lack of protocols and education) was reported as a facilitator if addressed at an organizational level. Interestingly, the collaboration with specially trained personnel was only reported as a facilitator possibly implying the complete unavailability of SLPs/SLTs and the anticipation by ICU teams to improve the care they offer to patients in the context of multidisciplinary collaboration. 

4.3.a Protocols and routine screening (lines 562-568)

4.3.b. Availability of screening and treating methods (lines 570-573)

4.3.c Barriers to standardized screening and treatment (lines 575-585)

4.3.d Facilitators to standardized screening and treatment (lines 587-593)

Discussion:

1. Overview and Objectives: The introduction to the discussion provides a clear overview of the study's objectives and the purpose of the survey.

However, it would be beneficial to explicitly state the main findings of the study to guide the subsequent discussion.

We addressed this comment by adding the following text in the in the first paragraph of the Discussion section:

The interpretation of the findings follows the same categorization as in the Results section. Overall, our results showed that a few ICU teams in Cyprus were aware of PED incidence in their units and most of them were aware of PED complications. Despite recognition of the need for evidence-based protocols as best practices for the screening and treatment of PED by most ICUs, very few routinely screened for dysphagia using appropriate methods. Similarly, protocols to guide PED management were not used in most ICUs, and effective treatments were limited by the lack of SLP/SPTs and/or knowledge gaps in ICU interprofessional teams.

Discussion (lines 377-384)

2. Perceived Best and Current Practices on PED: The discussion of the perceived best and current practices on PED is comprehensive and provides relevant insights. The comparison with findings from other studies adds value to the discussion.

However, it would be helpful to further elaborate on the implications of the absence of standardized dysphagia assessment protocols and the limited availability of SLP/SLTs in ICUs. Discussing the potential consequences of these factors on patient outcomes and the need for interdisciplinary collaboration would enhance the discussion.

This point was raised in:

Comment/ Results/ 2. Current Practices on PED/ a. PED Management

and in:

Comment/ Results/ 6. Perceived Best Practices on PED

Please, find our answer there.

3. Methods Used for PED Assessment: The discussion of the methods used for PED assessment adequately highlights the limitations of relying on certain assessment methods, such as the water swallow test, and the potential benefits of instrumental assessments like VFSS and FEES.

However, it would be beneficial to discuss the challenges and barriers to implementing these instrumental assessments in ICUs, such as the availability of technology and trained SLP/SLTs.

This point was raised in:

Comment/ Results/ 2. Current Practices on PED/ c. Methods Used for PED Assessment

Please, find our answer there.

4. Responsibilities of ICU Team Members: The discussion of the responsibilities of ICU team members in PED assessment is informative.

However, it would be valuable to further discuss the implications of relying on nurses and intensivists for PED assessment and the potential benefits of involving dedicated SLP/SLTs in the ICU team. Exploring the challenges and opportunities for expanding the role of nurses and implementing interdisciplinary collaboration could enrich the discussion.

We addressed this comment in the following section of the Discussion:

4.1.a.4 Responsibilities of ICU team members

In the majority of ICUs in our study, nurses and intensivists were responsible to assess PED as well as SLP/SLTs whereas in Switzerland nurses had the lead in the initial ICU dysphagia screening [9] and physicians in Germany [23]. None of the ICU teams that participated in our survey reported a dedicated SLP/SLT for ICU patients while approximately half of them reported a lack of SLPs/SLTs even as an external partner. In case of an available SLP/SLT on request, the percentage of patients consulted was less than 25%. It would be very interesting to investigate how many SLPs/SLTs are available at the moment in Cyprus for ICU consultations including PED management. Although the collaboration with an SLP/SPT in the ICU can positively affect ICU-related complications such as PED [44], the lack of SLP/SLTs involved in ICU patient care is common practice across the world [45].

What complicates the situation is that, according to the American Speech-Language-Hearing Association (ASHA), SLPs are the most qualified providers for dysphagia services and “cross-training of clinical skills is not appropriate at the professional level of practice" [46]. Yet, in some countries SLPs do not receive ICU-specific training, which may explain the tradition for lack of ICU-dedicated SLPs [47]. In the absence of SLP/ SLTs, PED identification has traditionally been performed by other healthcare professionals, mainly nurses. Nurse-performed dysphagia screening is considered to be feasible [26], safe [48] and superior to no screening in terms of patient outcome [49]. Until a dedicated SLP/SLT for PED screening and treatment becomes available for all ICUs, the empowerment of nurses through education along with the implementation of standardized protocols can contribute to the early identification of high-risk individuals for dysphagia and lead to referrals for optimal management. Apparently, professional and regulatory bodies of different health care professionals need to instigate interdisciplinary collaboration early in the education of undergraduate students which will hopefully lead to collaboration during clinical practice having patients as the point of reference.

4.1.a.4 Responsibilities of ICU team members (lines 437-464)

5. Prevention and Treatment of PED: The discussion of the interventions to prevent and treat PED is concise and provides relevant information.

However, it would be useful to discuss the limitations of the current interventions, such as muscle-strengthening exercises without swallowing, and the potential benefits of more comprehensive and targeted interventions. Addressing the challenges in implementing these interventions consistently across ICUs and the need for further research on effective treatment strategies would enhance the discussion.

This point was raised in:

Comment/ Results/ 3. Prevention of Aspiration Pneumonia Related to PED

and in:

Comment/ Results/ 4. Interventions to Treat PED

Please, find our answer there.

6. Awareness of PED Incidence and Consequences: The discussion of ICU team awareness of PED incidence and consequences is well-presented.

However, it would be beneficial to further discuss the potential reasons for the variation in awareness among ICU teams, such as differences in healthcare systems, interdisciplinary collaboration, and educational programs. Exploring strategies to improve awareness and knowledge among ICU teams could strengthen the discussion.

This point was raised in:

Comment/ Results/ 5. Scope of the Problem

Please, find our answer there.

7. The Discussion section provides a comprehensive overview of the survey findings and highlights important aspects related to the awareness, practices, and challenges of managing post-extubation dysphagia (PED) in ICUs.

However, it would be valuable to incorporate a paragraph discussing the potential implications of the results in the context of the COVID-19 pandemic.

We addressed this comment by adding the following text in the Conclusion section:

In the context of the COVID-19 pandemic and the associated respiratory failure and invasive ventilation, research findings suggest that dysphagia is prevalent in the population of critically ill patients with Covid-19, with 55-93% of patients experiencing swallowing difficulties [80,81]. The added challenge of COVID-19 further complicates dysphagia management as strict infection control measures, increased number of patients requiring ICU admission and limiting practitioners’ time and resources to care for all [82] may hinder comprehensive swallowing assessments and rehabilitative interventions. As a result, healthcare providers need to prioritize the early identification and management of dysphagia in critically ill COVID-19 patients to mitigate the potential complications and improve patient outcomes.

Conclusion (lines 663-673)

8. Any possible future roles of AI chatbots (like ChatGPT) in the future towards the individualized medical care of these ICU patients? (Examples: PMID: 37223340 and PMID: 37179277 )

We addressed this comment by adding the following text in the Conclusion section:

Looking the future of care of ICU patients, artificial intelligence (AI) chatbots like ChatGPT (OpenAI, San Francisco, CA, USA), can contribute to the individualized medical care of critically ill patients with dysphagia. By analyzing patient data (e.g. medical records, imaging scans and vital signs) AI can play a significant role in early detection of dysphagia, timely recognition of changes that may indicate worsening dysphagia or related complications, and risk assessment of complications associated with dysphagia. In this way, healthcare professionals can intervene promptly, implement appropriate treatment plans, and be alerted to take timely action. Additionally, by considering patient-specific factors like comorbidities, medication interactions, and swallowing function assessments AI can help develop individualized treatment plans for ICU patients with dysphagia.  AI can also recommend tailored interventions, including dietary modifications and rehabilitation exercises. As patients with dysphagia have to overcome communication barriers, AI-powered speech recognition systems can assist them in communicating their needs, difficulties, and discomforts more effectively, improving the overall care experience. Finally, AI can serve as a decision support tool for healthcare professionals by providing them with evidence-based recommendations, relevant research findings, and treatment guidelines specific to dysphagia management. This can assist clinicians in making informed decisions and improving patient outcomes [84,85].

It is important to note that while AI can provide valuable assistance, it should al-ways be used in conjunction with clinical expertise, skills, and knowledge of healthcare professionals to deliver the best possible care to ICU patients with dysphagia.

Conclusion (lines 680-700)

9. Limitations: The discussion of the study's limitations is appropriate and acknowledges the potential biases and limitations of the survey.

However, it would be helpful to further discuss the implications of these limitations on the generalizability and validity of the study findings, especially in the post-COVID era?

Exploring the potential impact of the limitations on the interpretation of the results and suggesting directions for future research would add depth to the discussion.

We made the suggested additions to the Limitation section (underlined texts):

First, this study presents a secondary analysis of an international cross-sectional survey conducted in 26 countries [27]. However, we still believe that our findings can provide a contribution to the existing literature as the context and environment of care can vary significantly among countries. Based on the above, regarding the results from Greek-Cypriot ICUs, this manuscript presented differences that are discussed in the discussion section compared to the findings of the DICE study [e.g The ICU teams in Cyprus seemed more aware of the risk of dysphagia to increase ICU and hospital stay (86% and 71% of respondents respectively) compared to the findings of the DICE study (64% and 42% respectively)]. Additionally, the results presented are more detailed. Finally, as, to the best of our knowledge, there is a small number of nationwide studies about the management of post-extubation dysphagia and awareness of best practices in ICUs (e.g. Swiss survey of dysphagia care [43], the MADICU study in Germany [23] and a Dutch national ICU survey [36]), reporting results from individual countries is important in informing quality improvement and education efforts. Not publishing these results would imply similar trends in a specific country, when in fact this is not the case.

Second, since the results were based on the perceptions of ICU teams around the incidence, risks and management of dysphagia in their units, the findings may not accurately reflect the actual prevalence and practices related to dysphagia management in critically ill patients.

Third, we did not collect data on the actual incidence and management of post-extubation dysphagia, as this could only be done prospectively due to lack of consistent documentation. This limitation implies that the study's findings may not capture the true scope and characteristics of dysphagia in critically ill patients.

Suggesting directions for future research, a prospective data collection, after ensuring that consistent and comprehensive documentation practices are in place to facilitate data collection, will provide more accurate and reliable information compared to relying on perceptions. Combining quantitative research with qualitative research methods such as interviews or focus groups, involving multiple ICUs across different regions or countries, will also help to enhance the generalizability of the findings and provide a more comprehensive understanding of dysphagia management practices in different healthcare settings.

Fourth, the absence of the national study coordinator during the questionnaire completion introduces the possibility of response bias. While senior nurses verified the completed questionnaires with the ICU team members, the lack of direct oversight raises concerns about the accuracy and consistency of responses. To minimize response bias in future research, having a study coordinator or research assistant present during the completion of questionnaires can ensure that the questions are understood correctly and provide an opportunity for immediate clarification if needed.

Finally, although we checked internal reliability and face validity of the questionnaire used, several other forms of reliability and validity were not performed, e.g. test-retest analysis, concurrent validity, etc. This limitation suggests that the robustness and accuracy of the study's findings may be compromised. Future research should in-clude test-retest analysis to evaluate the stability of the questionnaire over time and concurrent validity testing to establish its relationship with other relevant measures.

Overall, the limitations reported where the secondary analysis, the perception-based results, the lack of data on actual incidence and management, the response bias and questionnaire completion and the lack of comprehensive reliability and validity testing. These limitations highlight the need for caution when generalizing the study findings on the management of dysphagia in critically ill patients, especially to the post-COVID era where the landscape of critical care and ICU practices may have evolved, and new guidelines, interventions, or technologies may have emerged. In addition to the above suggestions for future research, further research, considering the unique challenges and changes brought about by the COVID-19 pandemic, would be necessary to provide more reliable and applicable insights into dysphagia management in critically ill patients in the post-COVID era, leading to improved patient outcomes and quality of care.

Limitations (lines 599-653)

The Conclusion section provides a concise summary of the study findings and emphasizes the need for improved awareness and knowledge regarding PED among ICU teams in Cyprus. It also highlights the importance of establishing protocols for the screening and treatment of dysphagia and emphasizes the severity of clinical complications associated with PED in ICU patients.

We thank the reviewer for this comment.

There are no significant English language issues. The manuscript is coherent, well-structured, and effectively conveys the main points of the research.

We thank the reviewer for this comment.

Reviewer 3 Report

This particular study is very interesting.

The manuscript is very well written, but it needs some small improvements, which I list below.

The introduction is very well written and contains everything the reader needs to understand the study.

Participants: detail how the participants were selected and what their entry and exclusion criteria were. Please also mention the final number of the sample.

Page 2 row 94-96 why the meeting was held with the participants and whether it was useful or not for the study

Data analysis: how the open-ended questions were analyzed

Page 4 row 160-161 it is not necessary to report the response rate of the units

Page 5 series 178-186 the information in this paragraph is very confusing please rewrite it

Page 9

Please clarify if the results reported refer to patients who tested positive for dysphagia or all patients

The discussion is well written.

 Τhe conclusions are well written.

Author Response

Reviewer 3

Reviewer’s Comment

Response

Citation (section and line/-s)

1. This particular study is very interesting.

We thank the reviewer for this comment.

2. The manuscript is very well written, but it needs some small improvements, which I list below.

We thank the reviewer for his comments which helped us to improve the feasibility of our manuscript.

3. The introduction is very well written and contains everything the reader needs to understand the study.

We thank the reviewer for this comment.

4. Participants: detail how the participants were selected and what their entry and exclusion criteria were. Please also mention the final number of the sample.

We addressed this comment by adding the following text:

…the participants were all the fourteen adult ICUs in the Republic of Cyprus designated for the management of critically ill, intubated patients. This was the final sample size of the study as there were no exclusion criteria.

Participants (lines 102-105)

5. Page 2 row 94-96 why the meeting was held with the participants and whether it was useful or not for the study

We addressed this comment by adding the following text:

…in order to enhance the accuracy of the data provided.

Participants (line 109-110)

6. Data analysis: how the open-ended questions were analyzed

This information was in 3.3.d Facilitators to standardized screening and treatment section.

We addressed this comment by adding the following text ‘Finally, the four open-type questions were not answered therefore they were not included in the analysis’ to the Data analysis section.

Data analysis (lines 185-186)

7. Page 4 row 160-161 it is not necessary to report the response rate of the units

We deleted this

8. Page 5 series 178-186 the information in this paragraph is very confusing please rewrite it

We rephrased this point as follow:

More than 85% (12 ICUs) of ICUs indicated that there was no standard protocol indicating which patients should be screened for PED. 9 ICU teams reported that no patient was screened for PED in their ICU. 2 ICUs reported that 51-75% of their patients screened for PED and in 1 ICU the percentage was 25-50% of patients. Screening for PED for less than 25% of their patients was reported in 1 ICU and only 1 ICU team reported that more than 75% of their patients screened for PED.

6 ICU teams reported that more than 75% of their patients who received a tracheostomy during their ICU admission screened for PED. 3 ICU teams reported that 51-75% of their tracheotomized patients screened for PED, in 3 ICUs the percentage was 25-50% of their patients and another 3 teams reported that no patient with tracheostomy was screened for PED in their ICU. Screening for PED after tracheostomy for less than 25% of patients was reported in 1 ICU.

Results

3.1.a. PED management (lines 208-219)

9. Page 9: Please clarify if the results reported refer to patients who tested positive for dysphagia or all patients

We thank the reviewer since this comment gives to us the opportunity to clarify this issue.

We made this addition to the Results section (underlined text):

…of aspiration pneumonia in all patients

3.1.b Prevention of aspiration pneumonia related to PED (lines 277-278)

10. The discussion is well written.

We thank the reviewer for this comment.

11. The conclusions are well written.

We thank the reviewer for this comment.

Round 2

Reviewer 1 Report

I previously reviewed this paper, and I got a revised version of it. I still have serious concerns regarding limitations of the generazibility of the paper.

Author Response

Reviewer 1

Reviewer’s Comment

Response

Citation (section and line/-s)

I previously reviewed this paper, and I got a revised version of it. I still have serious concerns regarding limitations of the generazibility of the paper.

We thank the reviewer for this comment which helped us to increase the rigor of our study.

We addressed this comment by adding the following underlined text in the Limitation section:

First, this study presents a secondary analysis of an international cross-sectional survey conducted in 26 countries [27]. So, an important limitation of the present work regards the fact that the findings are reported in relation to a particular healthcare setting, i.e., ICUs in the public and private sector of the Republic of Cyprus. It is important to note that ICU nurses in different healthcare systems across the nations may implement different protocols and care plans, or they may receive to various degrees continuing education on clinical guidelines. These concerns may influence the generazibility of the present results. However, the strength of the present study is the use of a nationwide representative sample of ICU nurses with a response rate of 100%, which i-creases the external validity of the study. Moreover, the present findings are based on a robust study design, which further supports the internal validity of the study.

Limitations, lines 576-585
